# Food odors trigger *Drosophila* males to deposit a pheromone that guides aggregation and female oviposition decisions

Chun-Chieh Lin[1], Katharine A Prokop-Prigge[2], George Preti[2,3], Christopher J Potter[1]*

[1]The Solomon H Snyder Department of Neuroscience, Center for Sensory Biology, Johns Hopkins University School of Medicine, Baltimore, United States; [2]Monell Chemical Senses Center, Philadelphia, United States; [3]Department of Dermatology, School of Medicine, University of Pennsylvania, Philadelphia, United States

**Abstract** Animals use olfactory cues for navigating complex environments. Food odors in particular provide crucial information regarding potential foraging sites. Many behaviors occur at food sites, yet how food odors regulate such behaviors at these sites is unclear. Using *Drosophila melanogaster* as an animal model, we found that males deposit the pheromone 9-tricosene upon stimulation with the food-odor apple cider vinegar. This pheromone acts as a potent aggregation pheromone and as an oviposition guidance cue for females. We use genetic, molecular, electrophysiological, and behavioral approaches to show that 9-tricosene activates antennal basiconic Or7a receptors, a receptor activated by many alcohols and aldehydes such as the green leaf volatile E2-hexenal. We demonstrate that loss of Or7a positive neurons or the Or7a receptor abolishes aggregation behavior and oviposition site-selection towards 9-tricosene and E2-hexenal. 9-Tricosene thus functions via Or7a to link food-odor perception with aggregation and egg-laying decisions.

*For correspondence: cpotter@jhmi.edu

**Competing interests:** The authors declare that no competing interests exist.

## Introduction

Animals must navigate a complex and changing environment for survival and reproduction. Odorants function as molecular cues for objects in the environment, and the olfactory system translates these cues into appropriate behaviors (*Suh et al., 2004*; *Laissue and Vosshall, 2008*; *Semmelhack and Wang, 2009*; *Stensmyr et al., 2012*). Living organisms are also a source of odorants, broadly termed pheromones, which play important roles in olfactory communications between different organisms of the same species (*Wilson, 1970*). Despite a wealth of knowledge of pheromone identities and their physiological functions, how environmental cues interact with pheromone signaling is not well understood.

A behavior largely mediated through pheromone signaling is population aggregation, which is hypothesized to ensure efficient use of resources (*Wyatt, 2014*). Aggregation behavior may reduce interspecific competition and also be important for finding mates (*Hedlund et al., 1996*). However, how aggregation pheromones are induced or deposited to mark certain geographical location and modulate animal behaviors remains largely unknown. In *Drosophila melanogaster*, aggregation behavior has been observed at locations that contain male flies and food substrates. The male specific pheromone cis-vaccenyl acetate (cVA) has been implicated as the key aggregation pheromone that attracts both males and females (*Bartelt et al., 1985*; *Xu et al., 2005*). cVA is manufactured and

**eLife digest** Animals rely on their sense of smell to navigate their environments; for example, the smell of food attracts animals to particular locations. These food-rich sites are also popular places for meeting, mating, and rearing offspring. Scent molecules emitted by animals can also attract others to a particular location or affect their behaviour. These molecules are known as pheromones.

Little is understood about how cues from food and pheromones interact to influence animal behavior. Studies of the *Drosophila* species of fruit fly have been conducted to tease out these interactions. Fruit flies are attracted to the smell of food—particularly overripe or rotting fruit—and often congregate at a food source to mate and lay their eggs. But whether it is the food itself or other cues that trigger these behaviors is not clear.

Now, Lin et al. reveal that male fruit flies emit a pheromone in response to the smell of food. This pheromone attracts females to the food to mate and encourages the females to lay their eggs at the food-rich site. This allows the male fly to have some say as to where his offspring will be laid and also increases the chances that his offspring will survive.

Using genetic and other experiments, Lin et al. found that the pheromone is detected by a receptor on the antennae of the female flies. This stimulates a specific type of brain cell that causes the female to lay her eggs at the site where the pheromone has been deposited. A chemical released by rotting fruit also stimulates these receptors and encourages the females to congregate and lay eggs.

The body of a male fly is coated by many different pheromones, yet he deposits only a select few upon smelling a food odor. How this occurs remains to be determined, but suggests that different pheromones might be localized to different body parts. By rubbing just those parts onto their surroundings, the male might be able to deposit a specific pheromone. How food odors specifically trigger this response, or if other flying insects also deposit pheromones in response to food odors, remains to be determined.

stored in an internal male organ (ejaculatory bulb) and transferred to females during copulation (*Brieger and Butterworth, 1970*; *Everaerts et al., 2010*), where it plays a role in inhibiting male courtship of previously mated females (*Ejima et al., 2007*). Low levels of cVA may be present on males prior to mating (*Bartelt et al., 1985*; *Farine et al., 2012*). Interestingly, flies defective in sensing cVA exhibit residual aggregation behavior, suggesting the existence of an aggregation compound besides cVA from male flies (*Xu et al., 2005*).

*Drosophila* pheromones are typically cuticular hydrocarbons that are produced by specialized cells (oenocytes) in the fly abdomen and form a waxy layer on the body surface (*Ferveur, 2005*; *Billeter et al., 2009*; *Wyatt, 2014*). Given the chemical nature of long chain hydrocarbons, most cuticular hydrocarbons are not volatile and are instead detected by gustatory contact (*Ferveur et al., 1997*). For instance, 7-tricosene, an abundant male cuticular hydrocarbon, functions as an aphrodisiac for females and anti-aphrodisiac for males and is sensed via the gustatory system (*Lu et al., 2012*; *Thistle et al., 2012*). Nonetheless, recent solid-phase micro-extraction gas chromatography experiments indicated the presence of volatile cuticular hydrocarbon pheromones, suggesting pheromone detection might be mediated through the olfactory system (*Ferveur et al., 1997*; *Farine et al., 2012*). All together, these studies suggest an uncharacterized cuticular hydrocarbon might function as an aggregation pheromone and possibly signal via the olfactory system.

Understanding how volatile pheromones may affect animal behavior is aided by identifying the odorant receptors activated by that pheromone. Studies using cuticular extracts or flies as odor sources identified neurons within trichoid sensilla as responding most robustly to *Drosophila* pheromones (*van der Goes van Naters and Carlson, 2007*). The pheromones activating each of the four pheromone receptors (Or67d, Or65a, Or88a, Or47b) have now been identified: cVA robustly activates Or67d (*Benton et al., 2007*; *Ejima et al., 2007*; *Kurtovic et al., 2007*; *Laughlin et al., 2008*) and, to a lesser degree, Or65a (*Ejima et al., 2007*; *Liu et al., 2011*); and fatty acid methyl ethers methyl laurate, methyl myristate, and methyl palmitate activate Or88a (*Dweck et al., 2015a*), whereas only methyl laurate activates Or47b (*Dweck et al., 2015b*). In each case, the pheromone receptor is specifically tuned to respond only to the identified pheromones, and exhibit little response to a large

panel of other odorants (*Hallem and Carlson, 2006*; *Dweck et al., 2015a*). In addition, the pheromones activate only the identified pheromone receptors, and show little effect on other olfactory receptors.

Here, we report the finding that male flies deposit an aggregation pheromone onto their surroundings upon apple cider vinegar odor and food odor stimulation. The pheromone, 9-tricosene, is a volatile male-specific cuticular hydrocarbon and requires the olfactory, but not gustatory, system for detection. By electrophysiological and behavior studies, we identify the Or7a olfactory neurons housed in basiconic sensilla as being necessary and sufficient for 9-tricosene pheromone detection. This is surprising as Or7a, in contrast to previously characterized pheromone receptors, can be classified as a 'generalist' odorant receptor as it can respond to many aldehydes and alcohols (*Hallem and Carlson, 2006*). Behaviorally, 9-tricosene promotes aggregation and modulates female oviposition site selection- a behavior that was previously considered a female exclusive decision (*Yang et al., 2008*; *Joseph et al., 2009*). The green leaf volatile E2-hexenal, a previously characterized robust Or7a agonist (*Hallem et al., 2004*; *Hallem and Carlson, 2006*), can also guide Or7a-dependent attraction and egg-laying decisions. Three additional Or7a odor agonists could also guide positive oviposition site selection, suggesting that Or7a neuron activation might directly influence this behavior. Indeed, selective optogenetic stimulation of Or7a neurons was also sufficient to guide egg-laying site selection. Our study provides important insights into biological communication by identifying an olfactory mechanism that links together food-odor perception, male pheromone deposition, species aggregation, and female oviposition decision-making.

## Results

### A novel chemosensory assay identifies a post-stimulus aggregation behavior

Traditional olfactory assays monitor either single flies (*Semmelhack and Wang, 2009*) or multiple flies in small spaces (*Quinn et al., 1974*) and might overlook important aggregation behaviors. We modified a four-field olfactory arena and fly tracking system (*Semmelhack and Wang, 2009*; *Ronderos et al., 2014*) to monitor large fly populations responding to odors over a large arena space. Flies are contained in a star-shaped arena between two glass plates (See Materials and methods for details), tracked in a dark chamber using infrared illumination (which is invisible to flies), and detected by an infrared camera (*Figure 1A*). We validated our experimental design by monitoring attraction to apple cider vinegar, repulsion to citronellal, and neutral responses to clean air (*Figure 1—figure supplements 1–3*). Flies mutant for *orco*, a necessary co-receptor for most olfactory receptor neurons (*Larsson et al., 2004*), showed reduced responses to control odorants, confirming the experimental design accurately assesses olfactory behaviors (*Figure 1—figure supplement 4*).

During investigations with the food-odor apple cider vinegar, we identified a novel olfactory behavior in which flies showed robust aggregation to the original odor quadrant for substantial time periods subsequent to odor application (*Figure 1B,C,G*). In these experiments, flies were stimulated with apple cider vinegar for 5 min, apple cider vinegar odor was switched to clean air for 10 min, and flies tracked in an arena that had been rotated 90° to rule out contamination in the odor delivery system (*Figure 1B*, *Videos 1 and 2*). Interestingly, this aggregation behavior in the absence of exogenous odor-stimulation persisted for >25 min (*Figure 1—figure supplement 5*). We called this behavior 'post-stimulus aggregation'. Perfusion of apple cider vinegar into the empty arena for 5 min in the absence of flies, and introduction of naïve flies into the arena, did not produce post-stimulus aggregation to the original quadrant (*Figure 1D*), suggesting the post-stimulus aggregation behavior is not due to residual apple cider vinegar on the glass plates.

### Post-stimulus aggregation involves pheromone deposition

Fly bodies are coated with cuticular hydrocarbons that can function as chemosensory pheromones (*Ferveur et al., 1997*; *Amrein, 2004*; *Ferveur, 2005*; *van der Goes van Naters and Carlson, 2007*). To rule out the possibility of passive pheromone deposition onto the glass plates due to crowding of many flies into a small space, flies were corralled into the odor quadrant by attraction to humidified air and monitored for post-stimulus aggregation behavior. Under these conditions, there was no detectable post-stimulus aggregation (*Figure 1E,F*, *Figure 1—source data 1*). All together, these data suggest that flies may deposit pheromone(s) in a specific response to the food-odor apple cider

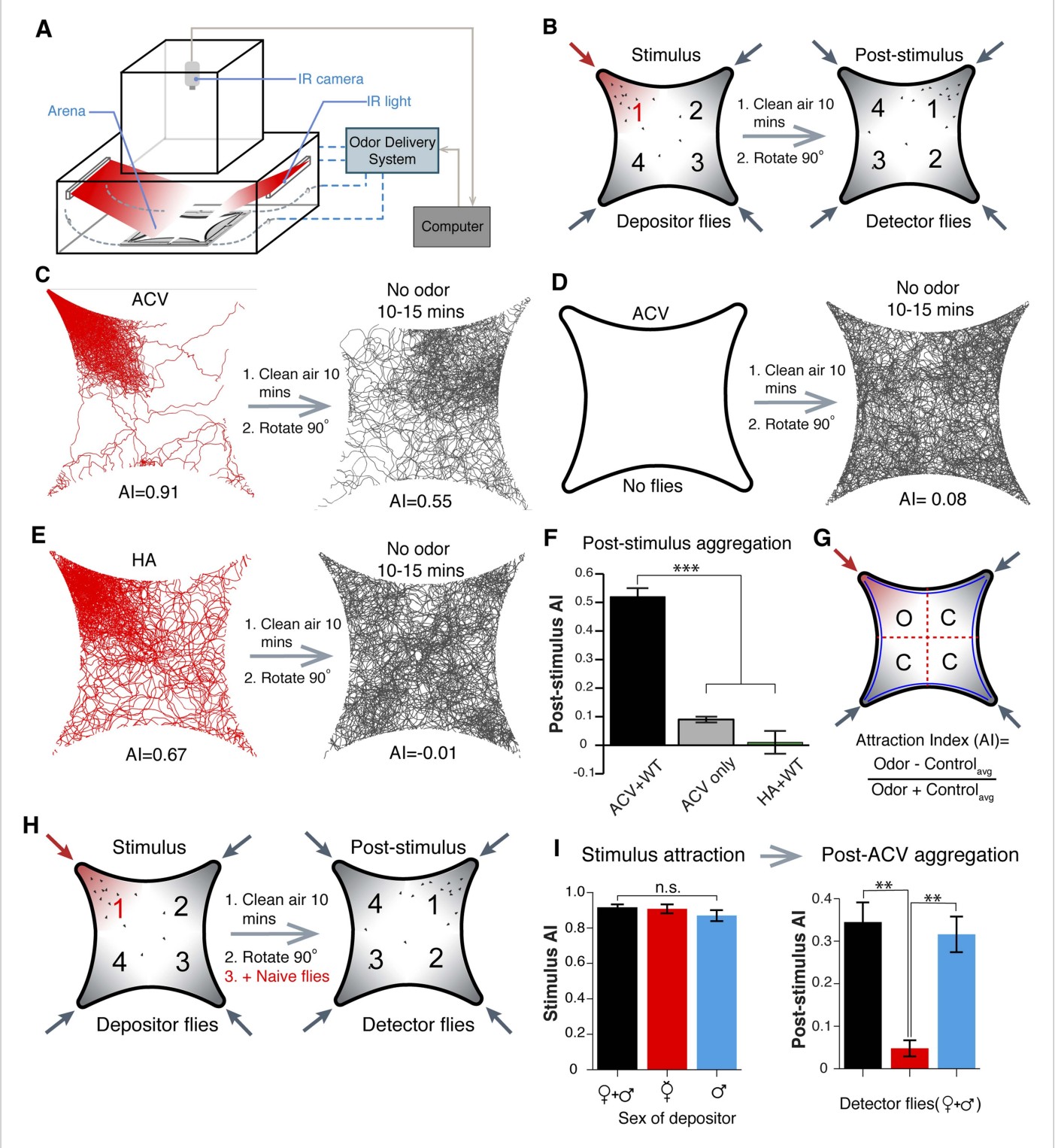

**Figure 1**. Identification of an apple cider vinegar odor induced post-stimulus aggregation behavior mediated by males. (**A**, **B**) Schematic of behavior setup and experimental design. (**C**) Fly tracking for 5-min of 25 male and 25 female wild-type flies. Flies are highly attracted to apple cider vinegar food odor, which gives rise to a post-stimulus aggregation behavior in the absence of exogenous odorants (right). (**D**) The lack of flies with apple cider vinegar stimulation (left) led to a lack of a post-stimulus aggregation (right). (**E**) Humidified air vs dry air is attractive (left), but does not lead to a post-stimulus aggregation (right). (**F**) Post-stimulus response summary (p = 0.0002 and 0.00001 for apple cider vinegar only and humidified air + WT, respectively; t-test, n = 3–6 per trial) (**G**) Definition of attraction index, A.I. Error bars indicate ±s.e.m. throughout. (**H**) Schematic of 4-quadrant arena using different

*Figure 1. continued on next page*

*Figure 1. Continued*

populations of depositor and detector flies. (**I**) Different depositor fly populations (females + males, females only, males only) were used as pheromone sources and assayed for post-stimulation aggregation by female + male detector flies (p = 0.003; t-test, n = 3–5 per combination).

The following source data and figure supplements are available for figure 1:

**Source data 1**. Source data for bar graphs shown in *Figure 1*.

**Figure supplement 1**. Basic characterization of the four-field olfactometer.

**Figure supplement 2**. Colormap of all fly trajectories from 0 min to 7 min.

**Figure supplement 3**. Four-field behavioral control experiments.

**Figure supplement 4**. Summary of four-field olfactometer controls.

**Figure supplement 5**. Time-course of aggregation pheromone responses.

**Figure supplement 6**. Post-stimulus aggregation induced by various concentrations of apple cider vinegar.

**Figure supplement 7**. Post-stimulus aggregation induced by additional food odors, but not by an attractive odorant.

**Figure supplement 8**. The aggregation pheromone is similarly attractive to virgin or mated males and females.

vinegar. Different concentrations of apple cider vinegar generated post-stimulus aggregation responses of different potencies (*Figure 1—figure supplement 6*).

## Post-stimulus aggregation behaviors are stimulated by food-odors

To determine if post-stimulus aggregation could be induced by other food odors, we tested the food odors banana and yeast paste (*Figure 1—figure supplement 7*). These stimuli were highly attractive to flies and also induced post-stimulus aggregation. As a further test, we examined stimulus attraction by a prominent attractive monomolecular odorant of apple cider vinegar: ethyl acetate. Although this odorant was highly attractive, it failed to generate post-stimulus aggregation behaviors (*Figure 1—figure supplement 7*). Together, these data suggest that post-stimulus aggregation behaviors may be guided specifically by food odors, and possibly by food odor perceptions. Future studies will be aimed at characterizing how food-odor perceptions might be encoded by a fly's olfactory system to direct pheromone deposition.

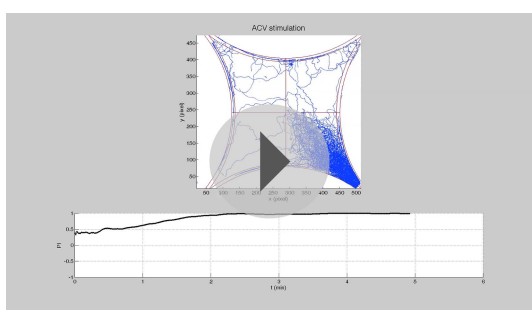

**Video 1.** Tracking flies stimulated with apple cider vinegar in four-quadrant behavior assay. Odor is supplied to the bottom right quadrant at time 30 s. The graph at the bottom reflects the attraction index over time. Related to *Figure 1*.

## Males are the source of the aggregation pheromone

Intra-species communications via pheromones are often sex-specific. We used different combinations of flies (mixed genders, virgin females, or males) as potential pheromone depositors upon apple cider vinegar stimulation, and new naïve mixed genders as detectors for the presence of the pheromone (*Figure 1H*). Only in the presence of male depositor flies did detector flies show post-stimulus aggregation (*Figure 1I*). This indicates that male flies are the source of the pheromone. The aggregation pheromone was equally attractive to both virgin and mated males and females (*Figure 1—figure supplement 8*).

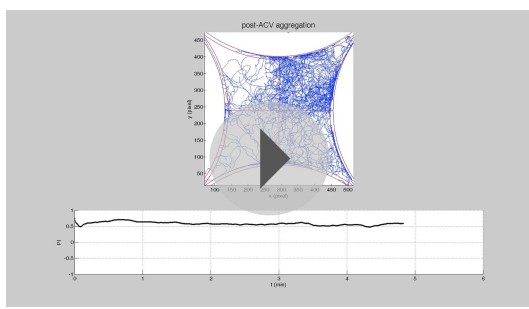

**Video 2.** Tracking flies responding to post-stimulus aggregation pheromone in four-quadrant behavior assay. The arena has been rotated 90° counter-clockwise with the aggregation pheromone deposits located at the top right quadrant. The graph at the bottom reflects the attraction index over time. Related to *Figure 1*.

## Detection of the aggregation pheromone requires *Orco*

Pheromones are detected by the olfactory and gustatory systems (*Amrein, 2004*; *Lu et al., 2012*; *Thistle et al., 2012*; *Wyatt, 2014*). Since the pheromone is deposited onto the glass plates, it might be detected by either chemosensory system. We utilized genetic mutants that are defective in specific modes of chemosensory signaling as detectors (*Figure 2A*). *Poxn* mutants, which exhibit no functional gustatory receptor neurons (*Awasaki and Kimura, 2001*), were as attracted as wild-type animals (*Figure 2B*, *Figure 2—source data 1*), suggesting that the gustatory system is not necessary for post-stimulus aggregation. Pickpocket channel 23 (ppk23), a Degenerin/epithelial sodium channel, is necessary for the detection of a male-predominant cuticular hydrocarbon, 7-tricosene (*Lu et al.,*

*2012*; *Thistle et al., 2012*). *ppk23* mutants exhibited similar post-stimulus aggregation compared to wild-type animals (*Figure 2B*), suggesting Ppk23 function is not necessary for aggregation and that 7-tricosene is unlikely to be the aggregation pheromone. Most insect olfactory receptors require a coreceptor(s) for normal olfactory responses: *Orco* for Odorant Receptors (*Larsson et al., 2004*) and *Ir8a* or *Ir25a* for ionotropic receptors (*Benton et al., 2009*; *Abuin et al., 2011*). The *Ir8a* and *Ir25a* double mutant flies exhibited normal post-stimulus aggregation, showing that most ionotropic receptors are not required for the pheromone attraction (*Figure 2C*). Interestingly, in *orco* mutant flies, attraction behavior to the pheromone was completely abolished and instead repelled by the apple cider vinegar quadrant (*Figure 2C*). This repulsion is likely due to acid sensing of minimal residual acetic acid on the quadrant mediated by the *Ir8a/Ir64a* complex (*Ai et al., 2010*) (*Figure 2—figure supplement 1*). Indeed, the *orco*, *Ir8a* double mutant was no longer repelled by the odor quadrant (*Figure 2C*), and the use of neutralized apple cider vinegar as the stimulus eliminated the post-stimulus repulsion demonstrated by *orco* mutants (*Figure 2—figure supplement 2*). We further tested if acidity of the apple cider vinegar was necessary for triggering wild-type post-stimulation behavior by neutralizing apple cider vinegar to pH = 7.0. No phenotypic difference was found compared to non-neutralized apple cider vinegar (pH = 3.2) (*Figure 2—figure supplement 3*). In sum, these data suggest that detection of the food-odor-induced pheromone is mediated through the *orco*-dependent olfactory system.

cVA has been suggested to be a male-derived aggregation pheromone in *D. melanogaster* (*Bartelt et al., 1985*). cVA induces conformational changes in the odorant binding protein LUSH, which enhances activation of Or67d/Orco complexes (*Xu et al., 2005*; *Laughlin et al., 2008*). Furthermore, the *Drosophila* CD36 homologue, sensory neuron member protein (Snmp), is essential for optimal Or67d neuronal activation (*Benton et al., 2007*). cVA can also activate Or67d/Orco complexes directly (*Gomez-Diaz et al., 2013*). Mutations of the key components in the signaling pathway (*Or67d*, *lush*, *snmp*) do not alter the post-stimulus aggregation behavior (*Figure 2—figure supplement 4*), suggesting that cVA is not the food-odor induced aggregation pheromone.

## 9-Tricosene is a food-odor induced aggregation pheromone

Since most insect pheromones are lipophilic carbohydrates dissolvable in hexane (*van der Goes van Naters and Carlson, 2007*), we reasoned that hexane might extract the active pheromone off the glass surface in the food-odor-induced quadrant. We induced wild-type flies to deposit the food-odor induced pheromone onto the glass plate and dissolved the deposited molecules into hexane. We then painted the hexane extract onto a new glass plate in a letter 'E' pattern. Naïve new flies were able to trace and follow the E pattern but do not follow control hexane extracts of flies stimulated by humidified air painted in the same pattern (*Figure 3A, B*, *Figure 3—figure supplement 1* and *Video 3*). These experiments demonstrated that an active pheromone component(s) was successfully preserved

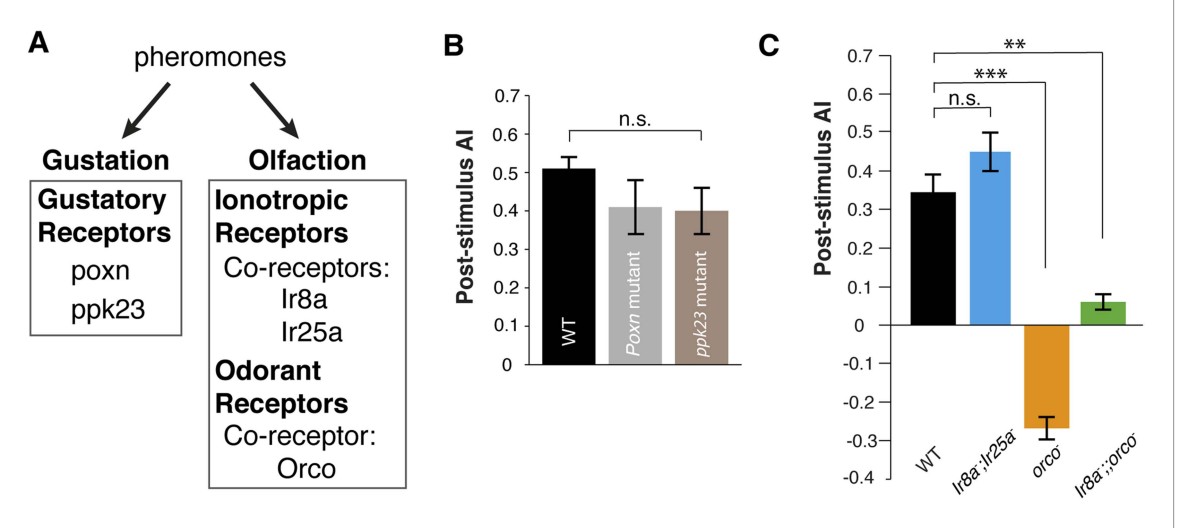

**Figure 2**. Post-stimulus aggregation requires Orco-dependent olfactory signaling. (**A**) Diagram indicating the different genetic components required for gustatory or olfactory-based pheromone detection. (**B**) Post-stimulus aggregation responses by gustatory receptor (*Poxn*; p = 0.2258; t-test) and *ppk-23* mutants (p = 0.0951; t-test, n = 4–5 per trail). (**C**) Post-stimulus aggregation responses by olfactory receptor (*orco*), and Ionotropic receptor (*Ir8, Ir25a*) mutants ($Ir8a^{-/-};Ir25a^{-/-}$: p = 0.1524; $orco^{-/-}$: p < 0.001; $Ir8a^{-/-};;orco^{-/-}$: p = 0.004; t-test, n = 4–6 per genotype). Wild-type flies were used as pheromone depositors.

The following source data and figure supplements are available for figure 2:

**Source data 1**. Source data for bar graphs shown in *Figure 2*.

**Figure supplement 1**. Ablation of the Ir64a + acid-sensing neurons increases post-stimulus attraction.

**Figure supplement 2**. Repulsion of *orco* mutant flies to the aggregation pheromone is likely mediated by acid sensing of residual apple cider vinegar.

**Figure supplement 3**. Acidity of apple cider vinegar is not required for post-stimulus responses.

**Figure supplement 4**. Mutating components of the cis-vaccenyl acetate pheromone pathway does not disrupt post-stimulus responses.

during the pheromone extraction. Behavioral results suggested the pheromone was volatile because: 1) it required the olfactory system for detection (*Figure 2C*); 2) constant air flushing reduced behavioral attraction after ~25 min (*Figure 1—figure supplement 5*); and 3) heating the pheromone-containing arena to 32°C to increase odor volatility eliminated post-stimulus behaviors (*Figure 3—figure supplement 2*). Recently, four cuticular hydrocarbons were identified as male-specific volatile pheromones: 7-docosene, 5-tricosene, 23-methyldocosane, and 9-tricosene (*Farine et al., 2012*). To identify the nature of the pheromone(s), we performed gas chromatography-mass spectrometry (GC–MS) analyses of hexane extracts from quadrants stimulated with apple cider vinegar alone, with humidified air + flies, and with apple cider vinegar + flies. Consistent with behavioral results that cVA is unlikely to be the food-odor induced pheromone (*Figure 2—figure supplement 4*), cVA was not detected in pheromone extracts from the glass plates (*Supplementary file 1*). 7-docosene and 23-methyldocosane were also not detected, while 5-tricosene was detected at trace amounts (*Figure 3C* and *Supplementary file 1*). The levels of 7-tricosene were increased in the experimental conditions. However, our behavioral results excluded 7-tricosene as the food-odor-induced pheromone (*Figure 2B*; also see below and *Figure 5—figure supplement 3*). Interestingly, only one other peak was significantly enriched in the experimental but not humidified air + flies control group: (Z) 9-tricosene (9-T) (*Figure 3C*, *Supplementary file 1*). Little is known regarding the function of 9-tricosene in *D. melanogaster* besides its presence as a male-specific volatile pheromone (*Everaerts et al., 2010*; *Farine et al., 2012*). To determine if 9-tricosene was attractive to *Drosophila*, as would be predicted for the food-odor induced aggregation pheromone, we used 9-tricosene as the

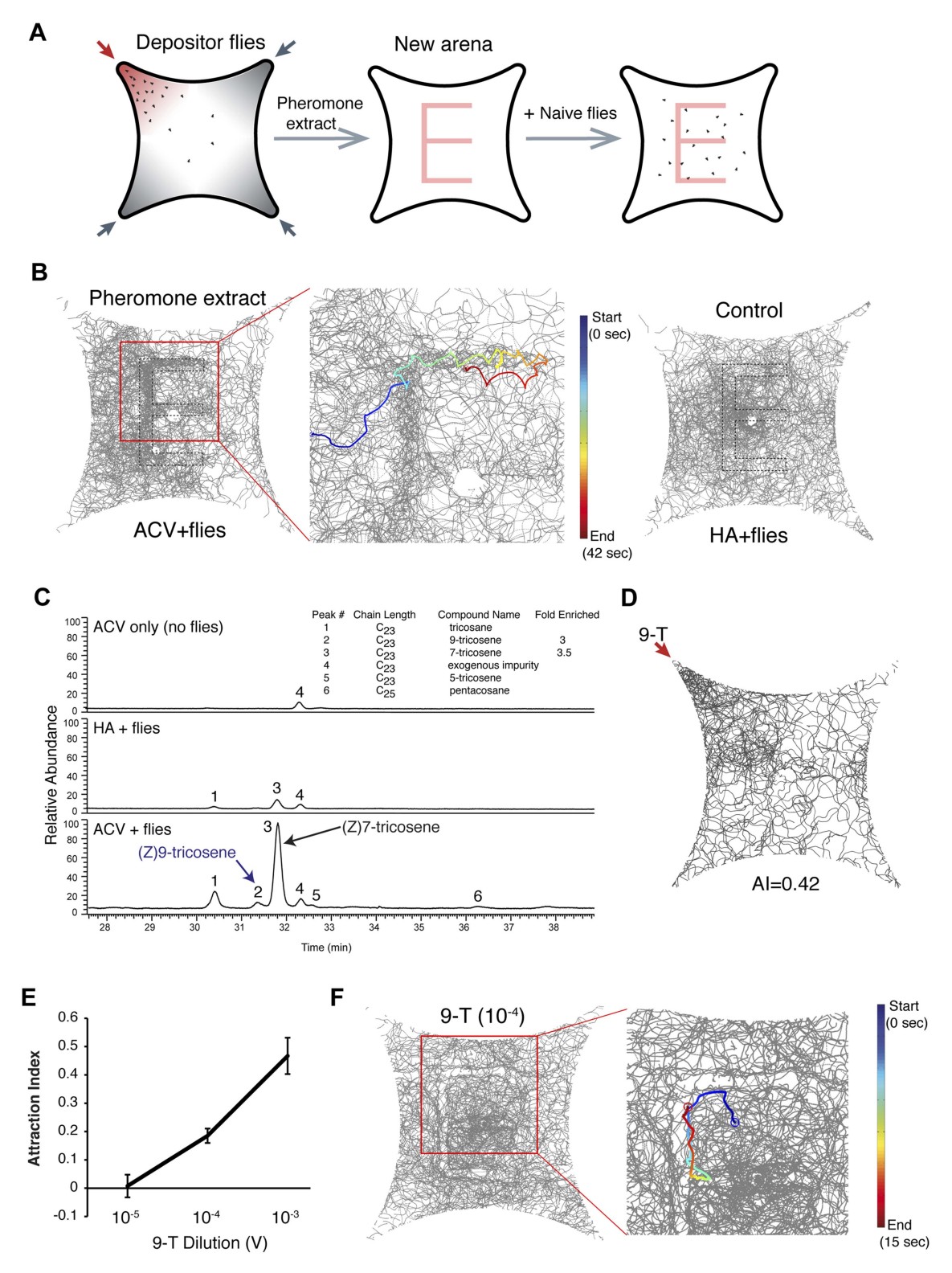

**Figure 3**. 9-Tricosene is a food-odor induced pheromone. (**A**) Schematic of pheromone extract paint experiment. (**B**) Hexane extracts of the pheromone quadrant were used to paint the letter 'E' onto the glass plate. Shown are traces of naïve new flies in the painted arenas by deposited pheromone extract (apple cider vinegar + flies) or control (humidified air + flies). The blue to red color trace indicates a single fly track from start to end of tracking. (**C**) GC–MS results of hexane extracts from quadrants stimulated by apple cider vinegar-only, humidified air and flies, and apple cider vinegar with flies. Peak #2 is

*Figure 3. Continued*

(Z)9-tricosene. 9-Tricosene exhibited a 2.8 fold enrichment on the glass plates upon food-odor stimulation. (**D**) Olfactory behavioral response of flies to 0.1% 9-tricosene. (**E**) Dose-response curve of 9-tricosene for mediating attraction. (**F**) Traces of flies in response to 9-tricosene deposited in an 'E' pattern on the glass plate. In this context, flies appear to be repelled by a concentrated 9-tricosene pattern and prefer to trail the 9-tricosene pheromone border. The behavioral differences between (**B**) and (**F**) maybe modulated by additional pheromone components present in the hexane extracts, or reflect that 9-tricosene trailing occurs only over a narrow odor range.

The following source data and figure supplements are available for figure 3:

**Source data 1**. Source data for *Figure 3E*.

**Figure supplement 1**. Single fly trajectories of painted 'E' experiment.

**Figure supplement 2**. The aggregation pheromone is heat-sensitive.

**Figure supplement 3**. Additional examples of flies responding to a 9-tricosene 'E' pattern.

**Figure supplement 4**. Apple cider vinegar stimulation of oenocyte-less males leads to a reduction in post-stimulus aggregation responses.

stimulus in the 4-field olfactory assay. Indeed, 9-tricosene elicits attraction behaviors (*Figure 3D-E*, *Figure 3—source data 1*). To determine if 9-tricosene could direct aggregation behaviors, we repeated the 'E' experiment using 9-tricosene alone (*Figure 3F*). Naïve flies did trace and follow the 9-tricosene pattern, paying particular attention to the odor border (*Figure 3—figure supplement 3*). Nonetheless, the aggregation response was not identical to the full hexane extract. In particular, flies appeared to be repelled by the center of a concentrated 9-tricosene odor trail. The sensory cause of this repulsion remains to be identified, but may arise from high concentrations of 9-tricosene mediating olfactory or gustatory contact repulsion. Such highly concentrated 9-tricosene deposits are unlikely to be found after food-odor-induced pheromone deposition in the four-field olfactory assay. Overall, these data suggest that 9-tricosene under certain conditions can function as an aggregation pheromone, although additional pheromone components may contribute to the full aggregation phenotype.

9-Tricosene is a member of cuticular hydrocarbons produced by oenocytes on the fly abdominal wall (*Billeter et al., 2009*; *Everaerts et al., 2010*). 9-Tricosene and other cuticular hydrocarbon components, but not cVA or methyl ethers pheromones, can be genetically eliminated by specifically ablating oenocytes (*Billeter et al., 2009*; *Dweck et al., 2015a*). Oenocyte-less males mixed with wild-type females no longer produced a post-stimulus aggregation behavior, suggesting that an oenocyte-derived cuticular hydrocarbon is essential for post-stimulus aggregation (*Figure 3—figure supplement 4*). Although other oenocyte-derived cuticular hydrocarbons may contribute to the aggregation phenotype, this is consistent with the identification of 9-tricosene as a food-odor induced pheromone.

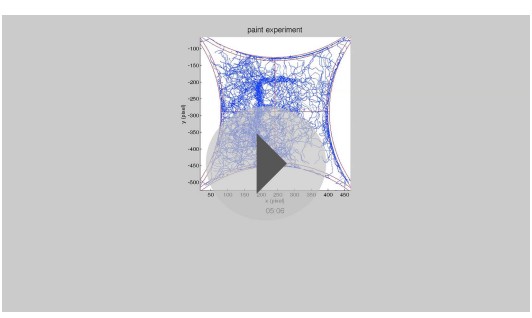

**Video 3**. Tracking flies responding to hexane extract of the post-stimulus aggregation pheromone painted as an 'E' pattern. Related to *Figure 3*.

## The Or7a receptor is necessary and sufficient for 9-tricosene activation

The olfactory sensory neurons in *Drosophila* can be classified as those that require either *orco* (odorant receptors expressed in basiconic, intermediate, trichoid and ac3 sensilla) (*Larsson et al., 2004*; *Couto et al., 2005*) or those that are *orco*-independent (ionotropic receptors expressed in coeloconic sensilla and gustatory receptors expressed in the ab1C neuron) (*Suh et al., 2004*; *Benton et al., 2009*; *Abuin et al., 2011*).

To determine which odorant receptors are required for 9-tricosene responses, we performed electroantennogram (EAG) recordings, which measure global detection of odor-induced antennal responses, in WT and orco mutants (*Figure 4A*). The orco mutants completely lacked responses to 9-tricosene (*Figure 4A,B*, *Figure 4—source data 1*). This suggests that 9-tricosene activates an orco-dependent odorant receptor, and does not require signaling from Ir or Gr receptors.

We next identified the orco-positive olfactory receptor neurons (ORNs) that respond to 9-tricosene, by using Fluorescence-guided Single Sensillum Recording (FgSSR) (*Lin and Potter, 2015*), which detects the activity of olfactory neurons within single sensory sensilla. Previously characterized volatile pheromones typically activate trichoid sensillar neurons (*van der Goes van Naters and Carlson, 2007*), and kairomones (odorants released by other animals or plants) typically activate intermediate sensillar neurons (*Stensmyr et al., 2012*; *Dweck et al., 2013*; *Ronderos et al., 2014*; *Dweck et al., 2015b*; *Lin and Potter, 2015*). Surprisingly, 9-tricosene did not stimulate these sensillar neurons (*Figure 4C*). We found that 9-tricosene elicits rapid and robust firing patterns in the antennal basiconic ab4 sensillum, which houses two neurons (ab4A and ab4B) that express Or7a (ab4A) or Or56a receptors (ab4B) (*Figure 4C*) (*Couto et al., 2005*; *Fishilevich and Vosshall, 2005*; *Stensmyr et al., 2012*). 9-Tricosene stimulates spiking of the larger amplitude neuron indicating the 9-tricosene-responsive ORNs are ab4A, which express Or7a receptors. Stimulation of ab4 sensilla by flies housed in a glass vial pre-stimulated with apple cider vinegar, compared to flies pre-stimulated with dry air alone, also led to significant increases in ab4A (Or7a) activation (*Figure 4—figure supplement 1*). The identification of Or7a as a 9-tricosene pheromone receptor was surprising as, unlike previously identified pheromone receptors, it had been shown to respond to a broad range of odors, including many aldehydes and alcohols (*Hallem et al., 2004*; *Hallem and Carlson, 2006*). To determine if the Or7a receptor is sufficient for 9-tricosene responses, we misexpressed Or7a in an olfactory neuron that lacks an odorant receptor in ab3A sensillar neurons (*Dobritsa et al., 2003*). Expression of Or7a endowed ab3A neurons the ability to respond to 9-tricosene comparable to the 9-tricosene activation pattern detected in endogenous Or7a-positive ab4 sensillum (*Figure 4D,E*). These data indicate that the Or7a receptor responds to 9-tricosene. This is unexpected since basiconic sensilla were traditionally considered food odor detectors (*Larsson et al., 2004*; *Suh et al., 2004*; *Fishilevich and Vosshall, 2005*; *Jones et al., 2007*; *Laissue and Vosshall, 2008*). Both male and female ab4A/Or7a neurons responded equally to 9-tricosene (*Figure 4E*), consistent with 9-tricosene being attractive to both males and females (*Figure 1—figure supplement 8*). Or7a neurons did not respond to cVA (*Figure 4—figure supplement 2*).

To further verify that Or7a was responsible for the 9-tricosene responses of ab4A neurons, we generated *Or7a* mutants using homologous recombination (*Figure 4—figure supplement 3*). Ab4 sensilla in *Or7a* mutants were identified based on their shape and the specific response of the ab4B neuron to geosmin (*Stensmyr et al., 2012*). No spontaneous or 9-tricosene stimulated ab4A spiking activity was observed in *Or7a* mutant flies (*Figure 4F,G*), indicating that Or7a is necessary for 9-tricosene activation in ab4 sensilla. All together, these data suggest that 9-tricosene specifically activates the 'generalist' Or7a receptor.

## The Or7a receptor is necessary for pheromone and 9-tricosene induced aggregation

To identify the full expression pattern of Or7a, we genetically converted our *Or7a* mutant to an *Or7a-GAL4* knock-in, in which *GAL4* is under control of the endogenous *Or7a* promoter (*Baena-Lopez et al., 2013*). *Or7a-GAL4* specifically drives effector expression only in olfactory neurons that target the DL5 antennal lobe glomerulus (*Figure 5A*) (*Couto et al., 2005*). We did not detect expression outside the antennae (*Figure 5—figure supplement 1*). We thus could use the *Or7a-GAL4* line to specifically ablate Or7a⁺ antennal neurons and test for changes in behavior. Aggregation responses to both the naturally deposited food-odor induced pheromone (*Figure 5B,C*, *Figure 5—source data 1*) and to 9-tricosene (*Figure 5D,E*) were completely abolished in Or7a neuron-ablated flies obtained using *Or7a-Gal4+UAS-hid* (*Figure 5*) or *Or7a-Gal4+UAS-DTI* (*Figure 5—figure supplement 2*). Ablation of other odorant receptors (*OrX-Gal4+UAS-hid*) did not affect aggregation (*Figure 5C*). Similarly, *Or7a* mutants completely lacked attraction to the naturally deposited food-odor-induced pheromone and to 9-tricosene (*Figure 5B-E*). To verify that aggregation behavior is specific to

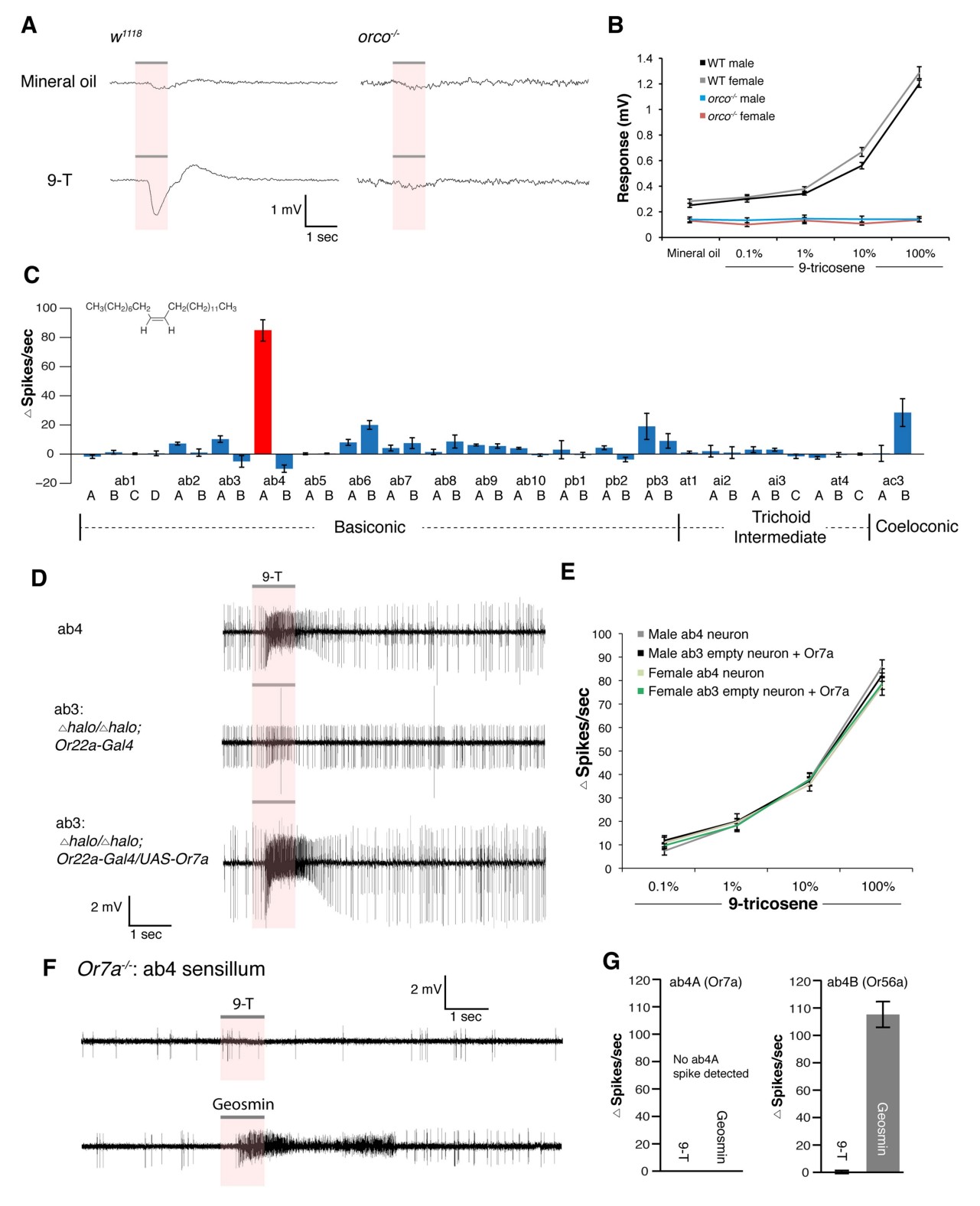

**Figure 4**. Electrophysiological results identify Or7a as the receptor for 9-tricosene. (**A**) Electroantennography (EAG) traces of wild-type and *orco⁻/⁻* flies stimulated with 100% 9-tricosene. (**B**) EAG response summaries of different 9-tricosene concentrations in different sexes of wild-type and *orco⁻/⁻* flies (n = 5–7 per stimulation). (**C**) Single sensillum recording (SSR) in all *orco*-positive antennal and maxillary palp sensilla. n = 3–6 per sensillum. (**D**) SSR traces showing responses to 9-tricosene stimulation in ab4 (9-tricosene responsive), ab3 empty neuron (*halo/halo;Or22a-Gal4*), and ab3 rescue (*halo/halo;*
*Figure 4. continued on next page*

*Figure 4. Continued*

*Or22a-Gal4/UAS-Or7a*) sensilla. (**E**) SSR response summary to 9-tricosene of native ab4 and rescued ab3 sensilla in different sexes. n = 7–8 per sensillum. (**F**, **G**) SSR trace responses and quantitative summary of ab4 sensillum of *Or7a* mutant flies stimulated with 100% 9-tricosene and geosmin (n = 4). Error bars indicate ±s.e.m. throughout.

The following source data and figure supplements are available for figure 4:

**Source data 1**. Source data for line and bar graphs in *Figure 4*.

**Figure supplement 1**. Fly odors can stimulate ab4A neurons.

**Figure supplement 2**. Response of ab4 sensillum to cVA.

**Figure supplement 3**. Generation of *Or7a* mutant.

9-tricosene, we repeated the experiments using the 9-tricosene pheromone isomer 7-tricosene (7-T), which contains an identical carbon chain length to 9-tricosene but a double bond at an alternate location. Interestingly, 7-tricosene induced a neutral to slightly repulsive behavior in WT flies (AI = −0.07 ± 0.026, *Figure 5—figure supplement 3*). These data suggest that aggregation behavior to the naturally deposited food-odor-induced pheromone depends on 9-tricosene and proper function of the *Or7a* receptor.

## 9-Tricosene guides oviposition site selection via Or7a neurons

Many behaviors occur at food sources, including courtship and egg-laying, but the molecular signals that help guide these behaviors remain poorly characterized (*Wyatt, 2014*). Since 9-tricosene may aggregate flies to sites of food-odor perception, we asked whether 9-tricosene could be an olfactory mechanism for catalyzing such behaviors.

*Drosophila* preferentially lay their eggs in food sources so as to increase survival of their progeny (*Joseph et al., 2009*; *Schwartz et al., 2012*; *Dweck et al., 2013*). Since 9-tricosene acts as a geographical marker for food, it could function as a male-generated guide for female egg-laying decisions. We modified our 4-field arena by spreading a thin layer of 1% agarose onto one of the glass plates to serve as a substrate for the deposited pheromone and an appropriate medium for female egg-laying (*Figure 6A*, *Figure 6—source data 1*). In order to rule out potentially confounding roles of males in this behavior, only previously mated females were assayed. Under conditions in which the food-odor-stimulated pheromone was deposited onto the agarose (*Figure 6A′*), females laid five-fold more eggs in the pheromone quadrant. This suggests that a deposited pheromone could guide female egg-laying site selection decisions. We next generated an arena in which one quadrant contained 9-tricosene (*Figure 6—figure supplement 1*). Female flies also laid significantly more eggs in locations containing only 9-tricosene (*Figure 6B′*). The 9-tricosene egg-laying preference was abolished when Or7a neurons were ablated (*Or7a-Gal4/UAS-hid* or *Or7a-Gal4/UAS-DTI*) (*Figure 6C′* and *Figure 6—figure supplement 2*). The 9-tricosene guided egg laying preference was also abolished in *Or7a* mutant flies (*Figure 6D*). The oviposition preference for the 9-tricosene quadrant was not due to the innate attraction to 9-tricosene because females spent similar time in the four quadrants over the course of the 23 hr egg-laying assay (*Figure 6E*).

Hydrocarbons could be potential food sources for larvae and female flies might thus preferentially lay eggs in locations containing cuticular hydrocarbons. To verify that oviposition guidance is specific to 9-tricosene, we repeated the oviposition experiments using the 9-tricosene pheromone isomer 7-tricosene (7-T), the most abundant cuticular hydrocarbon in male flies (*Everaerts et al., 2010*). Female flies did not preferentially oviposit in the 7-T quadrant (*Figure 6F*). Interestingly, total egg numbers laid were significant higher in *Or7a* mutant and Or7a neuron ablated flies, implying a potential connection of oviposition site selection and egg deposition number (*Figure 6G*). The female ovipositor can be involved in the detection and guidance to egg-laying cues (*Yang et al., 2008*). However, since Or7a is expressed only in the antennae and not in the female ovipositor (*Figure 5—figure supplement 1*), this implicates Or7a signaling in the antennae as the main driver

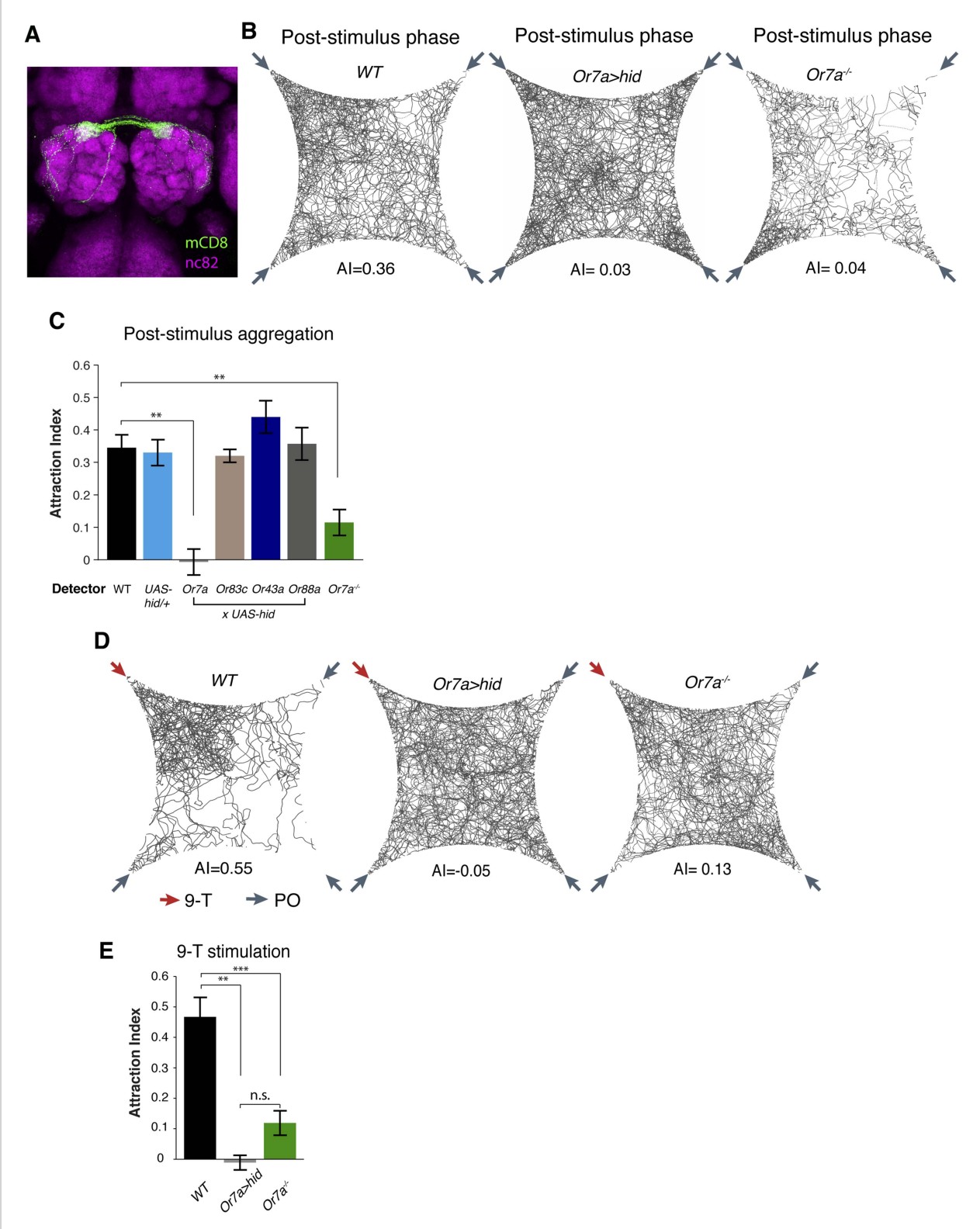

**Figure 5**. Or7a neurons are necessary for the behavioral response to naturally deposited aggregation pheromone and 9-tricosene. (**A**) Immunostaining of Or7a-expressing neurons innervating the DL5 glomerulus in the antennal lobe (*Or7a-Gal4/UAS-mCD8GFP*). (**B**, **C**) Four-field behavior responses of WT, *Or7a* mutant, Or7a-neuron ablated, and control OrX-neuron ablated flies (*Or83c, Or43a* and *Or88a-Gal4 x UAS-hid*) to naturally deposited pheromone

*Figure 5. continued on next page*

*Figure 5. Continued*

(p = 0.0012 and 0.006 comparing WT to Or7a neurons ablated and *Or7a*$^{-/-}$ flies; t-test, n = 4–6 per experiment). (**D**, **E**) Behavioral response of WT, *Or7a* mutant, and Or7a neuron ablated flies to 9-tricosene (0.1%) (p < 0.001; t-test; n = 4–5 per trial). Error bars indicate ±s.e.m. throughout.

The following source data and figure supplements are available for figure 5:

**Source data 1**. Source data for bar graphs in *Figure 5*.

**Figure supplement 1**. Whole-animal Or7a expression pattern.

**Figure supplement 2**. Or7a-ablation experiments for the deposited pheromone and 9-tricosene.

**Figure supplement 3**. Olfactory assays for 7-tricosene.

of Or7a-directed oviposition guidance. All together, these data suggest that male derived 9-tricosene can guide female egg-laying preferences, and this decision-making process requires proper antennal Or7a neuronal function.

## E2-hexenal activation of Or7a mimics 9-tricosene guided behaviors

Comprehensive SSR surveys of odorant–receptor activities identified multiple ligands for the Or7a receptor (*Hallem and Carlson, 2006*). E2-hexenal, a leafy green volatile released upon fruit or leaf damage (*Myung et al., 2006*), was identified as the most potent ligand for Or7a (*Hallem and Carlson, 2006*). E2-hexenal could thus be an abundant odorant for Or7a at damaged or rotting fruits and might contribute towards Or7a-mediated behaviors. We thus examined if E2-hexenal could direct similar behaviors as those triggered by 9-tricosene (*Figure 7*, *Figure 7—source data 1*). Indeed, we found that E2-hexenal, like 9-tricosene, directed Or7a-dependent egg-laying site selection (*Figure 7A-D*) and attraction (*Figure 7E*).

The presence of E2-hexenal, as a potentially abundant Or7a activator, might confound localized 9-tricosene guided behaviors. The concentration of E2-hexenal at a foraging site will vary depending on the fruit source, the stage of ripening (*Baldwin et al., 1991*), and the extent of fruit or leaf damage. Most undamaged plants emit undetectable levels of E2-hexenal (*Hatanaka and Harada, 1973*; *Bate and Rothstein, 1998*; *Farag and Pare, 2002q*; *Myung et al., 2006*). However, a single cut on an *Arabidopsis* leaf can produce ~28 parts per trillion of E2-hexenal (~$3 \times 10^{-11}$ E2-hexenal) (*Shiojiri et al., 2012*). As an upper limit of E2-hexenal concentrations at a foraging site, a fully disrupted source (*e.g*, a blended apple, tomato, or *Arabidopsis* leaf) can produce approximately 5–10 parts per million of E2-hexenal (~$10^{-5}$ -$10^{-6}$ E2-hexenal) (*Baldwin et al., 1991*; *Farag and Pare, 2002q*; *Myung et al., 2006*; *Chen et al., 2008*; *Shiojiri et al., 2012*). To determine if 9-tricosene pheromone cues remained recognizable in a surrounding of E2-hexenal, we repeated the 9-tricosene ($10^{-4}$ concentration) 3-choice oviposition-selection assays in the presence of high ($10^{-6}$, $10^{-7}$), middle ($10^{-8}$) or low ($10^{-9}$,$10^{-10}$) E2-hexenal concentrations (*Figure 7—figure supplement 1*). 9-tricosene functioned as an oviposition guidance cue in the presence of low and middle, but not high, E2-hexenal concentrations. These results suggest that 9-tricosene can effectively guide oviposition in the surrounding presence of E2-hexenal.

The ability of both 9-tricosene and E2-hexenal to guide oviposition-site selection suggested that Or7a activation might be a key signal for this behavior. We therefore tested the oviposition-site selection guidance to 3 additional Or7a-agonists (benzaldeyde, 1-butanol, 1-propanol) as well as 3 odorants that do not activate Or7a (hexyl butyrate, pentanoic acid, ethyl lactate) (*Hallem and Carlson, 2006*). Interestingly, all odorants that activated Or7a directed positive oviposition-site selection to some degree, whereas all odorants not activating Or7a showed mainly neutral or negative oviposition-site selection effects (*Figure 8*). The exception is pentanoic acid at high concentrations. This high concentration may recruit additional odorant receptors to positively influence site selection. Egg-laying site selection was associated with the specificity of the odorant for Or7a: those odorants (*e.g.,* benzaldehyde or 1-butanol) that activated many additional odorant receptors exhibited decreased oviposition towards that odorant whereas more specific Or7a agonists (*e.g.,* 9-tricosene and 1-propanol) demonstrated the highest positive oviposition site selection (*Figure 8—figure supplement 1*).

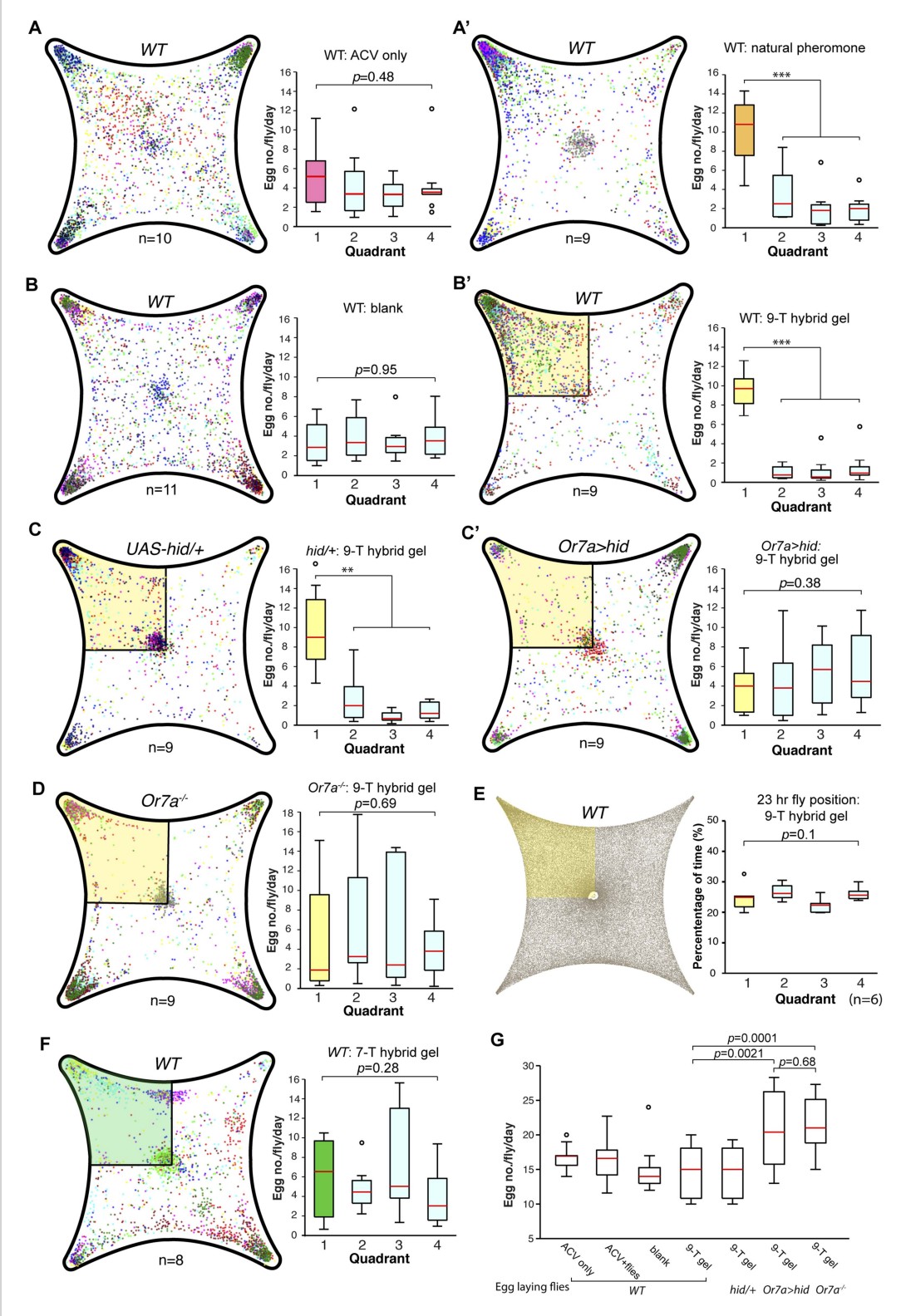

**Figure 6.** 9-Tricosene modulates female oviposition site selection. (**A**) Quantification and positions of eggs laid over ~23 hr in the 1% agarose arena with apple cider vinegar-only control or naturally deposited aggregation pheromone (A: p = 0.4836; n = 10; A': p < 0.001; n = 9; one-way ANOVA test) (**B**) Quantification and positions of eggs laid over ~23 hr in the agarose arena in blank control or with 9-tricosene (yellow, 0.001%) (B: p = 0.9499; n = 11; p < 0.001; t-test, n = 9 per trial, One-way ANOVA test). (**C**) The effect of 9-tricosene on female oviposition site selection was assayed in Or7a neuron
*Figure 6. continued on next page*

*Figure 6. Continued*

ablated flies (C: *UAS-hid/+*, p < 0.001, n = 9 per trial; C': *Or7a-Gal4/UAS-hid*, p = 0.384; n = 9, One-way ANOVA test). (**D**) 9-Tricosene guided oviposition site selection assayed in *Or7a⁻/⁻* mutant flies (p = 0.69; n = 9, One-way ANOVA test). (**E**) Positional recording throughout the 23 hr course of female oviposition behavior with a 9-tricosene hybrid gel (p = 0.1; n = 6, One-way ANOVA test). (**F**) Oviposition site selection using a 7-tricosene hybrid gel. (p = 0.28; n = 8, One-way ANOVA test). (**G**) Box plots indicating the total number of eggs laid in **A–D** (p = 0.0021 comparing WT and *Or7a > hid*, p = 0.0001 comparing WT and *Or7a⁻/⁻*, p = 0.68 comparing *Or7a > hid* and *Or7a⁻/⁻* ; t-test ; n = 9–11). In all panels, colored dots indicate actual egg locations. Different colors represent different experiment trials. Error bars indicate ±2.5 s.e.m. throughout. Data points not within this range are plotted as circles.

The following source data and figure supplements are available for figure 6:

**Source data 1**. Source data for box plots in *Figure 6*.

**Figure supplement 1**. Schematic of hybrid 9-tricosene gel construction.

**Figure supplement 2**. Oviposition guidance of Or7a-neuron ablated flies to 9-tricosene.

To further examine the sufficiency of Or7a neuron activation in guiding oviposition-site preference behaviors we utilized an optogenetics approach to specifically activate Or7a neurons with red light confined to one of the egg-laying wells (*Figure 8—figure supplement 2*). Since flies require feeding of all-trans-retinal for efficient light-induced activation of Channel Rhodopsin, we compared the behavioral responses elicited by red light to the same genotype of flies that were, or were not, fed all-trans-retinal (genotype: *Or7a-GAL4, UAS-CsChrimson*) (*Klapoetke et al., 2014*). Flies fed all-trans-retinal exhibited significantly increased egg-laying preference for the red light quadrant (*Figure 8—figure supplement 2*). These results support the findings that Or7a neuron activity can influence egg-laying decisions.

## Discussion

We have identified a phenomenon in which *Drosophila* males deposit the pheromone 9-tricosene in response to apple cider vinegar food-odor stimulation. This male-predominant cuticular hydrocarbon acts as an aggregation pheromone to attract both males and females and as a chemosensory cue to influence female oviposition site selection (*Figure 9*). The behavioral effects of 9-tricosene are mediated via specific activation of a broadly tuned odorant receptor. To females, activation of this odorant receptor imparts biologically relevant information regarding potentially beneficial egg-laying sites. This behavioral choice is likely modulated by the ensemble activity of other odorant receptors stimulated at the egg-laying site.

### Aggregation behavior guided by food-odor perception

Our results suggest that aggregation at a food-site might be strongly influenced by the olfactory perception of an optimal food source. We found that an aggregation pheromone is only deposited upon food-odor stimulations, such as apple cider vinegar, ripe banana and yeast. An attractive odorant component (ethyl acetate) found in apple cider vinegar was not sufficient for pheromone deposition, suggesting that attraction per se to an odor is not sufficient for aggregation pheromone deposition.

cVA has been suggested as the aggregation pheromone in *D. melanogaster*. However, in contrast to aggregation pheromones identified in other *Drosophila* species (*Bartelt et al., 1985*; *Hedlund et al., 1996*), cVA is only weakly attractive on its own (*Xu et al., 2005*), although does enhance aggregation when coupled with food or food-derived odors (*Bartelt et al., 1985*; *Everaerts et al., 2010*). However, mutating key components in the cVA signaling pathway did not affect the food-odor induced aggregation behavior. Moreover, GC–MS analysis of the active aggregation pheromone extract found no evidence that cVA was deposited. Therefore, cVA is likely not responsible for the aggregation behaviors triggered by food odors in these experiments. Nonetheless, it remains possible that under natural conditions, cVA is used as a long distance co-attractant with food odors at mating sites while 9-tricosene is used as a short-range aggregation pheromone at food sites.

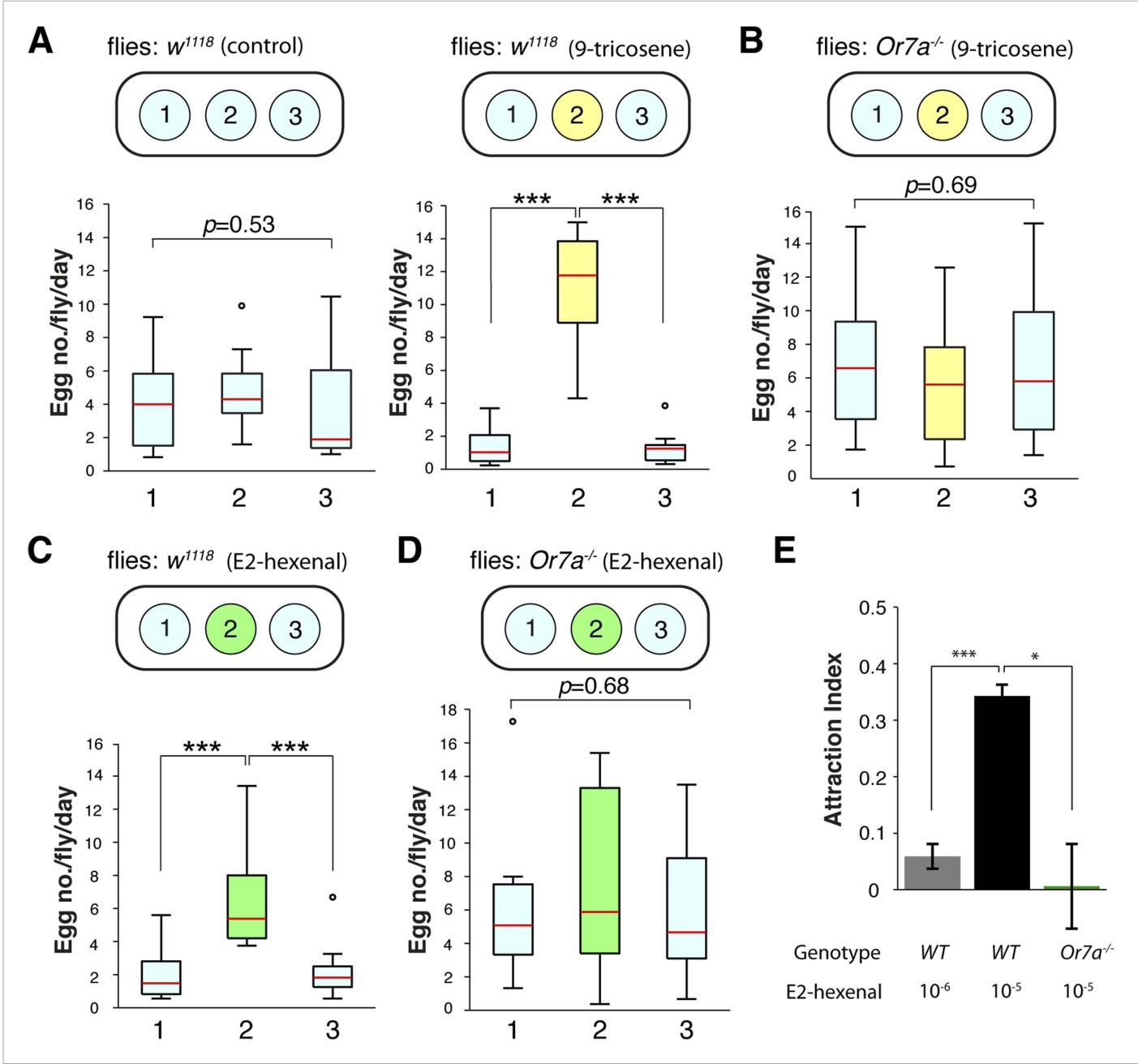

**Figure 7**. E2-hexenal modulates oviposition site selection. (**A**) The effects of 9-tricosene guidance on egg-laying using a 3-well spot plate (34 × 85 mm) containing 9-tricosene (10⁻⁴ dilution) in a 1% agarose gel (yellow, 0.001%; p < 0.001; t-test, n = 12) or control 1% agarose gel (blue) (p = 0.53, One-way ANOVA test, n = 17). (**B**) Egg laying preference of *Or7a⁻ᐟ⁻* mutant flies in the 3-well spot 9-tricosene egg laying assay (p = 0.69, One-way ANOVA test, n = 15). (**C**) Egg laying preference of *w¹¹¹⁸* flies in a 3-well spot egg laying assay (10⁻⁶ dilution) (p = 1.73×10-4, t-test, n = 10). (**D**) Egg laying preference of *Or7a⁻ᐟ⁻* mutant flies in a 3-well spot E2-hexenal egg laying assay (p = 0.68, One-way ANOVA test, n = 10). For box plots (**A–D**), error bars indicate ±2.5 s.e.m. Data points not within this range are plotted as circles. (**E**) Attraction of wild-type *w¹¹¹⁸* or *Or7a* mutant flies to E2-hexenal as determined in the four-field olfactory assay (p = 2.4×10⁻⁴ for comparing 10⁻⁶ and 10⁻⁵ E2-hexenal dilutions; p = 0.0149 for comparing *WT* and *Or7a⁻ᐟ⁻* at 10⁻⁵ E2-hexenal dilution; t-test,. n = 4–6 for each condition).

The following source data and figure supplement are available for figure 7:

**Source data 1**. Source data for box plots and bar graphs in *Figure 7*.

**Figure supplement 1**. Oviposition selection to 9-tricosene in a surrounding presence of E2-hexenal odors.

**A**
**Odorant to OR activites**

|  | Or7a | Or35a | Or67a | Or67c |
|---|---|---|---|---|
| E2hexenal | ++++ | ++++ | ++ | 0 |
| Benzaldehyde | ++++ | + | ++++ | 0 |
| 1-butanol | +++ | +++ | + | + |
| 1-propanol | +++ | + | 0 | 0 |
| Hexyl butyrate | - | ++++ | + | 0 |
| Pentanoic acid | 0 | 0 | ++++ | 0 |
| Ethyl lactate | 0 | 0 | 0 | ++++ |

**B**
**Oviposition preference**

|  | $10^{-4}$ | $10^{-3}$ | $10^{-2}$ | $10^{-1}$ |
|---|---|---|---|---|
| E2hexenal | -- | ++ | +++ | n.d. |
| Benzaldehyde | - | ++ | +++ | -- |
| 1-butanol | n.d. | + | ++ | ++ |
| 1-propanol | n.d. | - | + | ++++ |
| Hexyl butyrate | n.d. | - | -- | - |
| Pentanoic acid | -- | --- | ++ | n.d. |
| Ethyl lactate | -- | - | - | n.d. |

**Figure 8**. Odorants that activate Or7a guide oviposition site selection. (**A**) Summary graph of Or activities induced by 7 different odorants as detected by single sensillum recordings. All odorant responses are from (*Hallem and Carlson, 2006*). ++++, spikes ≥200; +++, spikes ≥150; ++, spikes ≥100; +, spikes ≥50; 0, spikes ≥0; -, spikes ≤0. (**B**) Oviposition-guidance preference for each odorant as assayed in the 3-choice assay. An oviposition preference index (OPI) was calculated as: (# of eggs laid in odor well – average # of eggs laid in control wells) / (# of eggs laid in odor well +average # of eggs laid in control wells) See *Figure 8—Source data 1*. ++++, OPI ≥0.8; +++, OPI ≥0.5; ++, OPI ≥0.2; +, OPI = 0–0.2; -. OPI = 0~-0.2; –, OPI ≤ −0.2; —, OPI ≤ −0.5. n = 5–9 for each odor concentration. n.d., not determined.

The following source data and figure supplements are available for figure 8:

**Source data 1**. Source data for values graphically represented in *Figure 8*.

**Figure supplement 1**. Oviposition preferences by different Or7a agonists and control odorants in a 3-well egg-laying assay.

**Figure supplement 2**. Optogenetic activation of Or7a neurons in egg-laying assay.

For many flying and non-flying insects, pheromone trails play an important role in organizing social foraging behaviors (*Schorkopf et al., 2007*; *Wyatt, 2014*). The first animals to find a suitable food source use a pheromone trail to recruit even more animals to the food source, which in turn deposit additional pheromone trails, and thereby establish a positive feedback of pheromone signaling towards the foraging site (*Schorkopf et al., 2007*; *Wyatt, 2014*). Our experiments identified pheromone tags deposited when *D. melanogaster* males perceived food odors, implicating the existence of such a behavioral occurrence in this insect. We found that 9-tricosene, and Or7a neuron activities, are likely key mediators of this particular pheromone response. Interestingly, the 'E' experiments with 9-tricosene suggests that 9-tricosene may cause repulsion at high concentrations, although it remains to be determined if these high concentrations represent natural conditions. Future experiments will investigate how 9-tricosene responses are modulated by other chemicals or pheromones in guiding pheromone trailing. It also remains to be explored if different pheromone trails are deposited under different environmental conditions.

## 9-Tricosene is an important close-range social pheromone

Food searching behaviors rely on volatile chemosensory cues, which create long-distance odor plumes or homogeneous odor clouds to guide foraging behaviors (*Budick and Dickinson, 2006*). E2-hexenal, because of its high volatility, might represent such a long-distance cue for recently fallen fruit or plants. Since E2-hexenal is generated by a plant-dependent enzymatic reaction, its levels will diminish over extended time periods (*Farag and Pare, 2002q*; *Myung et al., 2006*). In contrast, 9-tricosene, as a low volatility olfactory pheromone, might be ideally suited as a stable close-range olfactory marker and utilized to convey environmental conditions. This is supported by the variety of functions 9-tricosene exhibits in other species. For instance, 9-tricosene is a sex attractant for female houseflies (*Musca domestica*) (*Carlson et al., 1971*) and honey bee waggle dancers (*Apis mellifera*) use a pheromone cocktail that includes 9-tricosene to communicate with nest mates about the locations of

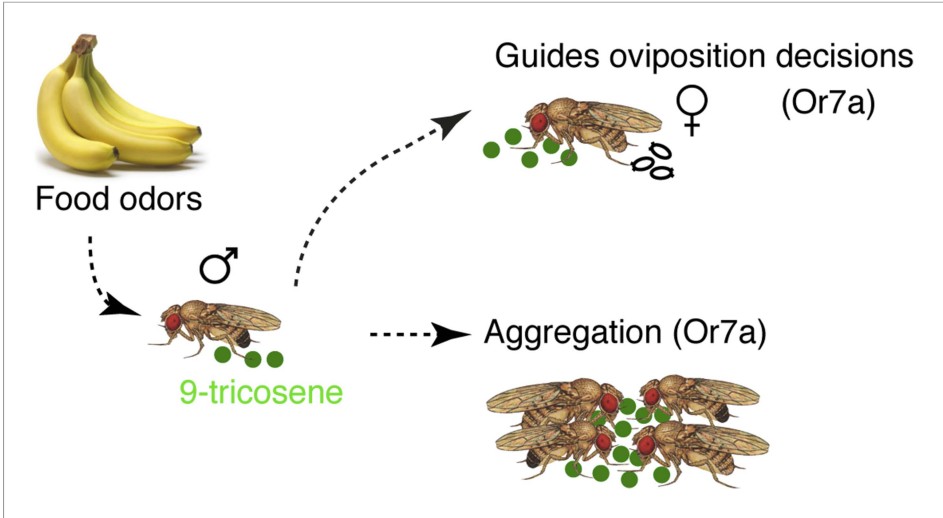

**Figure 9**. Model of food-odor induced pheromonal behavioral responses. Upon exposure to food odors, male *Drosophila melanogaster* deposit the pheromone 9-tricosene. 9-Tricosene functions via Or7a odorant receptors to guide aggregation and oviposition site-selection decisions. Activation of Or7a by other odorants may also guide similar behaviors.

food sources (*Thom et al., 2007*). Future experiments will be required to determine the ecological niche whereby 9-tricosene signaling in *Drosophila* is most prevalent.

## Pheromones can activate basiconic sensillar neurons

Four basic-types of olfactory sensilla have been classified based on their size, shape, and predicted functions (*Couto et al., 2005*). ORNs housed in basiconic sensilla tend to be strongly activated by food odors; ORNs in intermediate/trichoid sensilla tend to be activated by kairomones and pheromones; and ORNs in coeloconic sensilla are activated by amines, ammonia, water vapor and putrescine (*Hallem and Carlson, 2006*; *Benton et al., 2009*). Only two exceptions are known—phenylacetic acid is a food odor that activates coeloconic Ir84a + neurons (*Grosjean et al., 2011*); and carbon dioxide is detected by basiconic neurons expressing Gr21a/Gr63a (*Jones et al., 2007*). 9-tricosene represents the first *Drosophila* cuticular hydrocarbon pheromone that, to our knowledge, is demonstrated to activate a *Drosophila* basiconic sensilla. Intriguingly, the same sensilla and Or7a olfactory receptor responds to the silk moth pheromone bombykol (*Syed et al., 2006*). It is interesting to speculate that this *Drosophila* neuron's response to bombykol might represent off-target ligand specificity for 9-tricosene. Our data further suggests that activation of odorant receptors by both pheromone and plant volatiles might be more widespread than previously anticipated. Indeed, an olfactory receptor in the moth *Agrotis ipsilon* that was previously considered to be pheromone specific was recently found to also respond to the plant volatile heptanal, despite the plant volatile showing no structural similarity to the moth pheromone (*Rouyar et al., 2015*).

## Male pheromones influence a female oviposition decision

Oviposition site selection is a model system to study simple decision-making in *Drosophila* (*Yang et al., 2008*; *Joseph et al., 2009*). The behavior comprises three steps—an ovipositor motor program, a clean/rest period and a search-like behavior. The ovipositor motor program that leads to egg deposition is relatively short (6–7 s) as compared to clean/rest and search-like behaviors (100–130 s) (*Yang et al., 2008*). Rapid egg-laying associated with an extended positional search is consistent with our observations that over long time periods there were no detectable positional preferences in the 9-tricosene pheromone quadrant even though female flies preferentially laid eggs in this quadrant (*Figure 6*). These findings also suggest that 9-tricosene might mediate two temporally distinct responses in our experimental design. A short-term aggregation behavior that lasts ~25 min and a long-term oviposition site selection behavior that lasts for hours. A possible mechanism underlying

these different behaviors could be that detection thresholds for the two behaviors are different, that is, higher concentrations of 9-tricosene triggers aggregation whereas low 9-tricosene concentrations affect oviposition.

In many insects, eggs are vulnerable and larvae have restricted motility, thus oviposition site selection is a crucial decision for progeny survival. The hypothesis of 'mother-knows-best' stipulates that female egg-laying decisions evolved to oviposit in places offering the best survival of offspring (*Soto et al., 2014*). As expected, oviposition decisions require multiple sensory modalities, such as visual, olfactory, gustatory and proprioception (*Yang et al., 2008*; *Joseph et al., 2009*; *Schwartz et al., 2012*). Our study shows that a previously considered female-only decision can in fact be modulated by a male-deposited pheromone. Since 9-tricosene is enriched only upon food-odor stimulation, and acts to aggregate animals and increase courtship (Lin and Potter, unpublished observations), it could be a mechanism used by *Drosophila* males to increase the likelihood that their progeny will be laid in an optimal location. Thus, in addition to 'mother-knows-best', this suggest that 'father' may have co-opted a female's olfactory system in order to influence an egg-laying decision for maximizing progeny survival.

The identification of Or7a as a pheromone receptor represents an intriguing puzzle. 9-Tricosene is a specific activator of Or7a, but Or7a is not activated only by 9-tricosene. How can a specific pheromone response be mediated by a 'generalist' odorant receptor? Our behavioral experiments suggest a strong positive correlation with the specificity of Or7a activation and oviposition-guidance. This suggests that activation of the Or7a olfactory receptor neuron can strongly influence egg-laying decisions. 9-Tricosense may therefore have a specific influence on oviposition decisions due to its specific activation of Or7a. Or7a olfactory neurons join a growing list of olfactory neurons found to mediate specific olfactory behaviors like oviposition-site selection (*Dweck et al., 2013*, *2015b*), aversion (*Suh et al., 2004*; *Ai et al., 2010*; *Stensmyr et al., 2012*), courtship (*Kurtovic et al., 2007*), and attraction (*Semmelhack and Wang, 2009*). Odorants besides 9-tricosene that stimulate Or7a neurons will activate additional olfactory receptor neurons that may function to mask or modulate Or7a-mediated egg-laying behaviors. This likely reflects how olfactory systems function to make sense of a complex environment, by assigning biological weights and values to different olfactory neuron activity patterns that together influence a behavioral choice.

Food odors and pheromone signals have been shown to project to non-overlapping divisions in higher brain regions in *Drosophila*, suggesting that distinct brain divisions may be involved in mediating different biological functions (*Jefferis et al., 2007*). Might these disparate olfactory signals for oviposition decision converge in the female brain? ORNs expressing the same ORs converge onto the same glomeruli and synapse with second order projection neurons, which relay the olfactory information to higher brain regions (mushroom body calyx and lateral horn) (*Jefferis et al., 2007*). We demonstrate that basiconic Or7a neurons are responsible for 9-tricosene and E2-hexenal guided oviposition decisions (*Figure 6C,D,7C,7D*). Recently, it was found that flies preferred citrus fruit as oviposition substrates as detected by Or19a + olfactory neurons (*Dweck et al., 2013*), and also preferred to oviposit on substrates containing ethylphenols as detected by Or71a + olfactory neurons (*Dweck et al., 2015b*). Interestingly, the Or7a DL5, Or19a DC1, and Or71a VC2 projection neurons share highly similar axonal projection patterns in the lateral horn that are distinct from previously described food and pheromone regions (*Jefferis et al., 2007*; *Ronderos et al., 2014*). This suggests that oviposition site-selection might be strongly guided by a dedicated olfactory processing brain center.

## Materials and methods

### Experimental procedures

#### Fly stocks

Wildtype flies are isogenized $w^{1118}$ (IsoD1 $w^{1118}$) and IsoD1 $w^+$. Sources of the lines used in the study: *poxn* mutant, *ppk23⁻* (*Thistle et al., 2012*), *Ir8a⁻* (*Abuin et al., 2011*), *Ir25a⁻* (*Benton et al., 2009*), *Ir64a-Gal4* (*Ai et al., 2010*), *snmp1* mutant (*Benton et al., 2007*), *lush* mutant, constitutive active *lush* (*Laughlin et al., 2008*), *Or67d-Gal4* (*Kurtovic et al., 2007*), Δhalo/CyO; UAS-Or7a, Δhalo; Or22a-Gal4 (*Hallem and Carlson, 2006*), PromE(800)-Gal4, Tub-Gal80 ts/TM6B (*Billeter et al., 2009*), Or7a-Gal4

(BS#23907 (*Couto et al., 2005*)), *orco* mutant (BS# 23130 (*Larsson et al., 2004*)), *Or83c-Gal4* (BS#23910), *Or43a-Gal4* (BS#9974), *Or88a-Gal4* (BS#23138). Before any behavioral analyses were performed, mutant or transgenic stocks were backcrossed at least 5 generations to *IsoDI w¹¹¹⁸*.

## Imaging and immunohistochemistry

Confocal images were taken on a LSM 700 Confocal Microscope (Zeiss). The procedures for fixation, immunochemistry and imaging were as described previously (*Wu and Luo, 2006*). Primary antibodies used were Rat anti-CD8 (Caltag Laboratories, 1:200) and Mouse anti-nc82 (DSHB, 1:25).

## Four-quadrant behavioral assay

A four-quadrant olfactometer (*Vet et al., 1983*; *Semmelhack and Wang, 2009*) was used to track the olfactory responses of multiple flies at 30 frames/second (*Katsov and Clandinin, 2008*). Central air passed through a carbon filter before being split into multiple channels each regulated by a high-resolution flowmeter (Cole–Parmer). Electronically controlled 3-way solenoid valves (Automate Scientific, Berkeley, CA) regulated if clean air leaving the flowmeters expelled into the room or entered into custom made odor chambers (*Lundstrom et al., 2010*). Teflon tubing was used for odor delivery. The Teflon fly arena is 19.5 cm by 19.5 cm, with a thickness of 0.7 cm. Glass plates were secured onto the arena using clamps. The airflow of each quadrant was maintained at a rate of 100 ml min$^{-1}$ and verified by an electronic flowmeter before each experiment. Apple cider vinegar was diluted in $H_2O$ to make the final concentrations of 6.25% (1/16), 1.56% (1/64) and 0.39% (1/256) and acetic acid in water to make final concentration of 0.33%. Ethyl acetate and 9-tricosene (Sigma #859885) were diluted in paraffin oil for final concentration of 0.001% and 0.1%. When paraffin oil was used as solvent in the odor chamber, paraffin oil alone used in the three non-odor control chambers. 40–50 flies with an isogenized genetic background (*IsoD1 w¹¹¹⁸*) were used. At the time of the assay, flies were 4–6 days old and had been starved in vials containing 1% agarose for 40-42 hr to increase locomotor activity. The dark arena was illuminated by 2 infrared LED arrays (AL4554-880; Advanced Illumination, Rochester, VT), monitored by an infrared camera (Sony XC-EI50), and flies tracked by previously described software (*Katsov and Clandinin, 2008*). Data was analyzed by custom Matlab scripts. On average, each fly generates approximately 1800 tracked positional data points per min. If two flies intersect, their respective previously continuous tracks are considered completed, and new independent tracks begun once they move apart. This assures continuously labeled tracks originated from the same fly. An Attraction Index (AI) is defined as $(Ot_5-Cavgt_5)/ (Ot_5+Cavgt_5)$, in which $Ot_5$ is the number of tracked positional data points in the odor quadrant and $Cavgt_5$ is average number of tracked positional data points in non-odor control quadrants over a 5 min testing period. An AI = 1 indicates all flies were tracked to the odor quadrant, and an AI = 0 indicates flies were equally distributed to all four quadrants.

## Pheromone extraction

A 1:1 mixture of 40–44-hr starved 50 male:female flies were stimulated with humidified air or apple cider vinegar for 5 min to deposit substrates onto cleaned glass plates. Apple cider vinegar stimulation alone (without flies) was used as a negative control. Odors were followed by clean air perfusion for another 5 min. To generate a hexane pheromone extract, the glass plates were treated 3 × with 500 μl hexane solvent. The apple cider vinegar-stimulated pheromone extract was used to pipette a pattern onto a new clean glass plate or stored at −20°C for GC–MS analysis. The humidified air-stimulated hexane extract was used as a negative control in the hexane painting. In GC–MS experiments, to monitor extraction efficiency, 750 ng of internal standard controls (hexacosane (Sigma #241687) and triacontane (Sigma #263842) as dissolved in hexane) were added on to the glass plates immediately before pheromone extraction procedures. In the 9-tricosene 'E' paint experiment, 9-tricosene was dissolved in hexane solvent (1:25,000 dilution) and 150 μl was evenly pipetted onto the clean glass plate to form an 'E' pattern.

## Gas chromatography/mass spectrometry (GC/MS)

A sample volume of 2 μl of the hexane extract was injected in splitless mode into a Thermo Scientific ISQ single quadrupole GC/MS (Waltham, MA) with Xcalibur software (ThermoElectron Corp.) for separation and analysis of the deposited hydrocarbons. The GC/MS was equipped with a Stabilwax column, 30 m × 0.32 mm with 1.0 μm film thickness (Restek Corp., Bellefonte, PA). The injection port was set at 230°C. The oven temperature was set to 60°C, raised to 180°C at 6°C min$^{-1}$, held at 180°C for 20 min, and then raised to 220°C at 6°C min$^{-1}$ where it was maintained for an additional 20 min. Helium carrier gas constantly flowed at 2.5 mL min$^{-1}$. The mass spectrometer was operated at an ionizing

energy of 70 eV with a 2 scan/s rate over a scan range of $m/z$ 40–400 and an ion source temperature of 200°C. Identification of structures/compounds was performed using the National Institute of Standards and Technology library, as well as comparisons with known literature compounds and commercially available standards. Relative retention times were obtained by comparison of sample hydrocarbons to authentic standards. All standards were purchased from Sigma or Cayman Chemical Company at the highest available purity.

## Electroantennography (EAG)

Electroantennograms were recorded with capillary glass electrodes (1.5 mm outer diameter) containing *Drosophila* saline plus Triton X-100 (188 mM NaCl, 5 mM KCl, 2 mM $CaCl_2 \cdot 2H_2O$, 0.02% Triton X-100). The reference electrode was placed in the head capsule close to the base of the antenna. A polished large diameter (∼40–50 μm) recording electrode was capped onto the anterior distal region of the *Drosophila* third antennal segment. Control odorant stimulations (1% and 10% cVA; data not shown) were used to verify that the recording electrode was properly sealed onto the distal antenna, 30 μl of different dilutions of 9-tricosene in mineral oil on filter paper was used as the pheromone stimulus. Electrical signals were acquired with a Syntech Intelligent Data Acquisition Controller IDAC-4-USB and quantified by measuring the mV value at the greatest deflection in the EAG trace.

## Fluorescence guided single sensillum recording (SSR)

Recordings were performed as previously described (*Lin and Potter, 2015*). Sensillum of targeted ORNs was identified using green fluorescence signals by crossing *OrX-Gal4* to *15xUAS-IVS-mCD8GFP* (Bloomington Stock #32193) (*Pfeiffer et al., 2010*). Extracelluar activity was recorded by inserting a glass electrode to the base of the sensillum of 4–10 day-old flies. Signals were amplified 100X (USB-IDAC System; Syntech, Hilversum, The Netherlands) and inputted into a computer via a 16-bit analog-digital converter and analyzed off-line with AUTOSPIKE software (USB-IDAC System; Syntech). The low cutoff filter setting was 50Hz, and the high cutoff was 5 kHz. Stimuli consisted of 1000 ms air pulses passed over odorant sources (*Dobritsa et al., 2003*). The Δspikes/second is obtained by counting the spikes in a 1000ms window from 500 ms after odor stimuli were triggered, subtracting the spikes in a 1000ms window prior to stimulation. Furthermore, the response generated by control solvent was further subtracted. The formula is as below:

(odor response-spontaneous response)-(solvent response-spontaneous response).

9-Tricosene from three different sources (Sigma Cat#859885, TCI America Cat#T1242, AK Scientific Cat#M691) robustly activated ab4A neurons. 9-tricosene from Sigma was used in all reported experiments. For SSR experiments using fly body odors, 50 male and 50 female flies were starved in vials containing 1% agarose gel for 40 hr as described above and then transferred to a 40-ml glass vial (Thermo scientific B7999-6). Dry air or apple cider vinegar (6.25%) was perfused through the vial for 30 min (flowrate = 100 ml/min). The vials were then used as odor sources to stimulate the ab4 sensilla in the standard SSR setup as described above. Each odor vial was used less than 3 times to avoid depletion of deposited odors.

## Generation of Or7a transgene

5′ and 3′ homology arms of the *Or7a* gene were generated by PCR amplifying from bacterial artificial chromosome (C.H.O.R.I, RP98-39F18) and *WT* genomic DNA, respectively and subcloned into the pTV^Cherry vector (*Baena-Lopez et al., 2013*). 5′ homologous sequence immediately 5′ to the ATG start site of Or7a (A of ATG is included) (4199 b.p.) was subcloned between NheI and KpnI restriction sites. A 4304 b.p sequence starting from 1368 base pairs downstream to the ATG start codon of Or7a was cloned between SpeI and BglII sites. In-Fusion cloning was used for subcloning into the pTV vector (Clontech Laboratories, Inc.) (*Figure 4—figure supplement 3*).

Primers used for PCR (Vector specific sequence in red, Or7a specific sequence in blue; lowercase letters indicate designed b.p. to preserve restriction sites):

5′ homology arm: 5′Or7a_FOR, GCT ACC GCG GGC TAG cCA ACA TGC CGA TTA TGT CG; 5′Or7a_REV, AGT TGG GGC ACT ACG gta ccT GGC TGA TGG ACT TTT GAC G

3′ homology arm: 3′Or7a_FOR, CGA AGT TAT CAC TAG tAG CCA AGT TCT CGT TTT CGC; 3′Or7a_REV, TTA TGC ATG GAG ATC tTT TGG CAT TGT GTG TTG CAC

## Generation of Or7a deletion mutant and GAL4 knockin

Accelerated homologous recombination was performed according to Baena-Lopez LA et al. (*Baena-Lopez et al., 2013*). Briefly, P-element insertion lines containing the Or7a knockout construct were

crossed to *hs-Flp, hs-SceI* (BS#25679) and heat-shocked at 48 and 72 hr after egg-laying (1 hr duration each time). Female progeny with mottled eyes were crossed to *ubi-Gal4[pax-GFP]* (*Baena-Lopez et al., 2013*) in order to select against flies containing non-homologous recombination events. Stocks were generated from candidate flies that contained both $w^+$ and GFP markers. *Or7a* mutants were verified by single sensillum recordings and PCR (*Figure 4F, G*, *Figure 4—figure supplement 3*). In order to identify the ab4 sensillum, 30 µl of geosmin (Sigma #16423-19-1), an odor that specifically activates only ab4B (Or56a) (*Stensmyr et al., 2012*), was used (*Figure 4F,G*).

Primers used for verification: G4polyA_FOR: TCG ATA CCG TCG ACT AAA GCC; gOr7a_REV:TCG CCG TTG AGT TTT CAG AG

The Or7a-Gal4 knockin was generated by co-injection of the pRIV-Gal4 donor plasmid (*Baena-Lopez et al., 2013*) with PhiC31 integrase to target GAL4 to the attP site within the knockout locus, as described in (*Baena-Lopez et al., 2013*).

## Four-quadrant egg laying behavior assay and positional recording

The schematic of the hybrid 9-tricosene or 7-tricosene gel is as shown in *Figure 6—figure supplement 1*. Control agarose gel was made by pouring 70 ml of a 1% agarose gel onto a glass plate assembled onto the 4-field arena. 9-tricosene or 7-tricosene gel was made by mixing 0.8 mg pure 9-tricosene (Sigma #859885) or 7-tricosene (Cayman Chemical Company #9000313) into 70 ml of 1% agarose gel (temperature = ~50℃). One quadrant of the control gel or 9-tricosene or 7-tricosene gel was cut out and transferred to a Petri dish. In order to increase egg production, mixed population of male and female flies were pre-induced in vials with wet yeast paste (yeast +0.5% propionic acid) overnight. Mated females from premixed population were rapidly separated by cooling on ice and transferred to the arena gel. The egg laying behavior was performed in a dark enclosure at room temperature for 22–23 hr. Simultaneous recordings of the fly positions were performed using the same setup for tracking of the four-quadrant behavior assay described above except that the frame rate was set at 1 frame/5 s due to the large file size generated over the extended time period. On average, each fly generated approximately 16,560 tracked positions per experiment. The recorded data was analyzed with custom Matlab programs and analyzed for the AI as defined above. Analyses scripts are available as *Source code 1* or upon request.

## 3-well egg-laying behavior assay

A schematic of the 3-well (34 × 85 mm) egg-laying assay is shown in *Figure 7*. The 9-tricosene well was made by mixing 1 µl 9-tricosene into a 70 ml 1% agarose gel solution. 1.2–1.3 ml of this solution was allowed to set in a well for 10 min before the experiment. Control wells contained only 1% agarose. Odorant wells (E2-hexenal (Sigma W256005), benzaldeyde (Sigma 418099), 1-butanol (Sigma 281549), 1-propanol (Sigma 279544), hexyl butyrate (Sigma W256803), pentanoic acid (Sigma 240370), and ethyl lactate (Sigma W244007)) were made by mixing 1 µl of $10^{-1}$ to $10^{-6}$ odorant concentrations into a 70 ml 1% agarose solution. Odor concentrations listed in Figures or Figure legends are the approximated final diluted concentrations in the agarose gel. Females were pre-stimulated as described above, placed in the 3-well agarose spots, covered with Petri dish lids (Diameter = 90 mm, Fischer Scientific FB0875712), and allowed to oviposit in a dark temperature controlled chamber for 23 hr. The oviposition preference index (OPI) is defined as $(E_o-E_{avg})/(E_o + E_{avg})$, in which $E_o$ is the number of eggs in the odor well and $E_{avg}$ is average number of egg numbers in non-odor wells. An OPI = 1 indicates all eggs were laid in the odor well, an OPI = 0 indicates eggs were equally distributed to all 3 wells, and an OPI = −1 indicates all eggs were laid in the non-odor wells.

## Channel rhodopsin activation

Newly eclosed flies were transferred to fly vials containing 0.4 mM of all-trans-retinal (Sigma–Aldrich #R2500, dissolved in pure ethanol) in fly food. The flies were fed with all-trans-retinal food in the dark for 3 days and used for the 3-well egg-laying assay as described above. A 627 nm LED light source (1-up LED Lighting Kit, PART #: ALK-1UP-EH-KIT; LEDSupply.Com) was placed directly beneath the glass 3-well egg-laying chamber with a white acrylic diffuser in between (Cat# 8505K11, McMaster-Carr). The LED was powered and controlled by an Arduino Uno to activate *UAS-CsChrimson* (*Klapoetke et al., 2014*) (Bloomington Stock # 55,135). The power supplied to the LED by the Arduino was set to 2 V with light on for 8 ms and off for 17 ms by using a custom Arduino program.

The red light stimulus was supplied continuously throughout the 23 hr egg-laying assay. The Arduino program is available as *Source code 1*.

## Acknowledgements

We thank R. Benton, J, Carlson, B Dickson, J Levine, D Smith, G Suh, K Scott, J-P Vincent and Bloomington *Drosophila* Stock Center (NIH P40OD018537) for fly lines. We thank R Reed, A Kolodkin for comments on the manuscript and K-W Yau, D Bergles, C Montell and M Wu for discussion. We thank XJ Gao, JJ Luo and O Riabinina for custom Matlab programming. We are grateful to D Ronderos for initial SSR experiments. This work was supported by grants from the Whitehall Foundation (CJP) and NIH NIDCD (R01DC013070, CJP).

## Additional information

### Funding

| Funder | Grant reference | Author |
|---|---|---|
| National Institute on Deafness and Other Communication Disorders (NIDCD) | R01DC013070 | Chun-Chieh Lin, Christopher J Potter |
| Whitehall Foundation | | Christopher J Potter |

The funders had no role in study design, data collection and interpretation, or the decision to submit the work for publication.

### Author contributions

C-CL, Conception and design, Acquisition of data, Analysis and interpretation of data, Drafting or revising the article; KAP-P, Acquisition of data, Analysis and interpretation of data; GP, Conception and design, Analysis and interpretation of data; CJP, Conception and design, Analysis and interpretation of data, Drafting or revising the article

## Additional files

### Supplementary files

• Supplementary file 1. Volatile male-specific and male-enriched pheromones detected by GC–MS under control and experimental conditions.

• Source code 1. Source Matlab and Arduino scripts for fly tracker analyses and optogenetic stimulations.

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
