## [Decision Letter]

Thank you for submitting your work entitled “Food odors trigger *Drosophila* males to deposit a pheromone that guides aggregation and female oviposition decisions” for peer review at *eLife*. Your submission has been favorably evaluated by a Senior Editor, a Reviewing Editor (Mani Ramaswami), and two reviewers.

The reviewers have discussed the reviews with one another and the Reviewing Editor has drafted this decision to help you prepare a revised submission.

In “Food odors trigger *Drosophila* males to deposit a pheromone that guides aggregation and female oviposition decisions,” Lin et al. report that food odors trigger male fruit flies to deposit an aggregation pheromone that also influences female oviposition preference. This phenomenon is in itself novel and interesting. Previous work on aggregation and oviposition has focused on how females are attracted by other females (leading to oviposition in patches, which may provide the progeny an advantage as they can better cope with competition from fungi and other fly species). This work is the first of its kind to show that males can drive female aggregation.

Using elegant multidisciplinary approaches, including behavioral assays, GCMS chemical analysis, electrophysiological recording and fly genetics, the authors identified 9-tricosene as a likely aggregation pheromone released by male flies in response to food odors and Or7a as the olfactory receptor that detects the pheromone. In support of this, the authors demonstrate that the attractive factor can be extracted in hexane and they postulate that the main factor is 9-tricosene on the basis of a GC/MS analysis. They demonstrate then that 9-tricosene (9-T) is detected by sensilla expressing Or7a. Ablating neurons expressing Or7a suppresses the attraction induced by 9-T (delivered as an odorant) as well as oviposition induction. Finally, they compare the effects of 9-T and E2-hexenal since E2-hexenal also activates Or7a.

Their findings demonstrate an intricate relationship between food odor stimulation and a cascade of behavioral responses that are beneficial to the survival of the species. This work is novel, interesting and important. The data and science are of high quality and the conclusions, if clearly established, will be of great interest to a broad range of scientists.

However, there are several issues that the authors absolutely need to address in a revised manuscript. One overriding issue and suggestion is to simplify the manuscript (potentially with a focus on the role of 9-T) as, in current form, it is unnecessarily long, convoluted and redundant, and leaves open too many questions. Another is that it suffers from inconsistent assays (which need to be addressed experimentally) as well as an unnecessarily selective presentation and discussion of their findings.

Essential revisions:

1) The biological activities found with the surface extracts are not strictly reproduced by 9-T. Thus other chemicals may be involved. Actually, the chemical analysis made by the authors shows that the extracts contain a higher proportion of 7-tricosene (7-T). The role of 7-T is not at all discussed nor tested leaving open the possibility that it is the mixture 7-T + 9-T (and possibly other molecules at a lower concentration) that are needed to recapitulate the biological effect of the whole extract. While 9-T seems to have an effect as an odorant, the nice “E” effect of the biological extract may need a contact chemical like 7-T or a chemical not detected through their chemical analysis.

2) On this note, the original behavior involved odorants (and tastants, which are argued against by the *poxn* experiments) left over a surface by males, while in subsequent tests using 9-T, the authors did not “paint” the surface with 9-T, but applied it as a volatile. These are different assays. Their “E” experiment (Figure 2) is very nice. The authors should duplicate it with 9-T. if this does not work then the authors should nevertheless present these negative data and intellectually engage with it in a revised discussion.

3) As reported by Hallem and Carlson, 2006, Or7a is one of the most broadly tuned odorant receptors. It responds strongly to many odorants, including several alcohols and aldehydes. Or7a even responds to bombykol, a moth sex pheromone (Syed et al., PNAS, 2006). As it is, the authors present their findings as if Or7a responds strongly only to E2-hexenal and 9-tricosene. There are three suggestions for improvement here:

A) Current data show that Or7a neurons are necessary. Additional experiments should ask if they are sufficient.

B) The authors should consider testing the behavioral effect of a number of different Or7a ligands to see whether 9-T is indeed a special, biologically relevant ligand for this behavior.

C) A revised discussion should clearly acknowledge and integrate previous findings. It is puzzling that a pheromone is detected by a generalist Or. However, even if the authors do not have a good answer, it is still a surprise and therefore an issue that should be highlighted so that all olfactory biologists think about it.

4) The authors emphasize that only “food odors or complex odors” could trigger the deposition of 9-tricosene. However, they did not provide sufficient experimental support to define the chemical or biological meaning of “complex food odors”. Which types of ORNs respond to the complex food odors? How many different types of ORNs are required to trigger the release of 9-tricosene? These are not trivial questions and deserve to be addressed fully in a separate study. The current manuscript will benefit significantly if the authors could remove some panels in Figure 1 and Figure 2 and combine these two figures to focus on the key finding that a food odor (ACV) triggers the release of an aggregation pheromone from male flies.

5) Similarly distracting is the molecular evolution analysis shown in Figure 8. They conclude their study by looking at the evolution rate of olfactory receptors in 9 *Drosophila* species and find that *Or7a* is a gene that underwent many amino acid substitutions, suggesting that this receptor is one of the most variable amongst different species and thus one of the least conserved.

The authors seem to imply that 9-tricosene/Or7a interaction is unique for *Drosophila melanogaster*. However, without testing whether 9-tricosene also activates ab4A or triggers similar aggregation or oviposition behavior in other *Drosophila* species, their molecular evolution analysis alone is insufficient and their interpretation appears forced. If the authors feel very strongly about making this point in the current study, they will need to conduct similar electrophysiological and behavioral experiments in other *Drosophila* species. Otherwise, it may help to streamline the work by removing Figure 8.

[Editors' note: further revisions were requested prior to acceptance, as described below.]

Thank you for submitting your work entitled “Food odors trigger *Drosophila* males to deposit a pheromone that guides aggregation and female oviposition decisions” for peer review at *eLife*. Your submission has been favorably evaluated by a Senior Editor, a Reviewing Editor (Mani Ramaswami), and two reviewers.

The reviewers have discussed the reviews with one another and the Reviewing Editor has drafted this decision to help you prepare a revised submission.

All of the reviewers felt that the manuscript had been greatly improved in revision and gained in clarity and coherence. However, they had two concerns that need to be addressed, ideally through specific additional experiments or, if these prove technically difficult, through necessary changes to the text.

1) Based on Figure 3 and Figure 3, it appears that the manuscript is incorrect in attributing the effect they observed with the pheromone blend to 9-T alone. The main reason is that the “E” distribution of the flies does not recapitulate the initial pattern and that the subsequent tests are different in essence from the initial tests. Figure 3 show clear attraction, while Figure 3 appears to show contact repulsion.

Two possible reasons for this should be tested:

A) The difference could be linked to the concentration of 9-T. This could be addressed by testing different concentrations – lower or higher of 9-T. Ideally, the authors should calculate the amount of 9-T in the pheromone blend used in 3B and repeat the 3F experiment with a similar 9-T quantity.

B) It is possible that there is an interplay between olfactory and gustatory modalities. Is it possible that one is olfactory (3D) and the apparent contact repulsion to 9-T is gustatory (3F)? This can be determined by testing the taste mutants (*ppk23* and *poxn*) for their response to 9-T alone (as in 3F) as has been been done in Figure 2 for the natural pheromone blend. The same experiment with *Or7a* mutants will also be informative.

The Discussion should be modified in accordance with the results of these experiments, if necessary to acknowledge that the effect of 9-T “alone” might be different than of 9-T + 7-T (which is really the dominant chemical in this blend).

2) The discussion on the role of the generalist Or7a could be improved and made more balanced.

---

## [Author Response]

[…] However, there are several issues that the authors absolutely need to address in a revised manuscript. One overriding issue and suggestion is to simplify the manuscript (potentially with a focus on the role of 9-T) as, in current form, it is unnecessarily long, convoluted and redundant, and leaves open too many questions. Another is that it suffers from inconsistent assays (which need to be addressed experimentally) as well as an unnecessarily selective presentation and discussion of their findings.

Essential revisions:

1) The biological activities found with the surface extracts are not strictly reproduced by 9-T. Thus other chemicals may be involved. Actually, the chemical analysis made by the authors shows that the extracts contain a higher proportion of 7-tricosene (7-T). The role of 7-T is not at all discussed nor tested leaving open the possibility that it is the mixture 7-T + 9-T (and possibly other molecules at a lower concentration) that are needed to recapitulate the biological effect of the whole extract. While 9-T seems to have an effect as an odorant, the nice “E” effect of the biological extract may need a contact chemical like 7-T or a chemical not detected through their chemical analysis.

In the original manuscript, we did check for a potential involvement of 7-tricosene in the aggregation phenotype (revised manuscript subsection “Detection of the Aggregation Pheromone Requires Orco”). We found that mutants of *pickpocket23* (the primary 7-tricosene receptor) or *poxn* (which converts gustatory sensory neurons to mechanosensory neurons) retained wild-type responses to the aggregation pheromone (revised Figure 2). In addition, the egg-laying preference phenotype of the naturally deposited pheromone is recapitulated by 9-tricosene but not by 7-tricosene (revised Figure 6). This strongly suggests that 7-tricosene, and/or other pheromones that primarily stimulate gustatory neurons, are not required for the aggregation pheromone response or oviposition guidance.

*2) On this note, the original behavior involved odorants (and tastants, which are argued against by the* poxn *experiments) left over a surface by males, while in subsequent tests using 9-T, the authors did not “paint” the surface with 9-T, but applied it as a volatile. These are different assays. Their “E” experiment (*Figure 2*) is very nice. The authors should duplicate it with 9-T. if this does not work then the authors should nevertheless present these negative data and intellectually engage with it in a revised discussion.*

We have duplicated the “E” experiment using 9-tricosene, and the results are now included as new Figure 3, and new Figure 3—figure supplement 3. We found that flies do track and aggregate to 9-tricosene deposits, but that the aggregation phenotype is not identical to the full aggregation pheromone (compared Figure 3 to Figure 3). This suggests that other (likely olfactory) components may modulate the full aggregation phenotype, but that 9-tricosene can function as an aggregation molecule.

3) As reported by Hallem and Carlson, 2006, Or7a is one of the most broadly tuned odorant receptors. It responds strongly to many odorants, including several alcohols and aldehydes. Or7a even responds to bombykol, a moth sex pheromone (Syed et al., PNAS, 2006). As it is, the authors present their findings as if Or7a responds strongly only to E2-hexenal and 9-tricosene. There are three suggestions for improvement here:

A) Current data show that Or7a neurons are necessary. Additional experiments should ask if they are sufficient.

To determine if activation of Or7a neurons are sufficient to guide oviposition-site selection preferences, we took an optogenetics approach to activate Or7a neurons in the presence of red light (wavelength 627 nm; genotype: *Or7a-GAL4; UAS-CsChrimson*). Flies require feeding of all-trans-retinal (ATR) for optogenetic activation of channel rhodopsins, and so we compared the same genotype of flies that were, or were not, fed ATR. We found that activation of Or7a neurons could lead to a significant increase in egg-laying preference towards substrates illuminated by the stimulating red light. The significance of this experiment is reduced by a wide range of behaviors of the -ATR control flies to the red-light source, which might arise from experimental conditions outside our control. Nonetheless, these results do suggest that activation of Or7a neurons is sufficient to drive oviposition-site selection. Since these results essentially confirm a chemogenetics approach to Or7a activation described below, we have included the optogenetics experiments as new Figure 8—figure supplement 2.

B) The authors should consider testing the behavioral effect of a number of different Or7a ligands to see whether 9-T is indeed a special, biologically relevant ligand for this behavior.

We have now examined the egg-laying behavioral effect to 3 additional Or7a ligands (benzaldeyde, 1-butanol, 1-propanol), as well as 3 odorants that do not activate Or7a (hexyl butyrate, pentanoic acid, ethyl lactate). Pentanoic acid and hexyl butyrate were selected because they specifically activate Ors that benzaldehyde and 1-butanol also activate: Or67a and Or35a. We found that all Or7a ligands could, to differing degrees, induce positive egg-laying preferences. In contrast, all non-Or7a ligands showed neutral or negative egg-laying preferences. Interestingly, odorants that more selectively activated Or7a (e.g., 1-propanol) showed the greatest preference in egg-laying assays. These results suggest that activation of Or7a neurons can strongly influence positive egg-laying site preferences. 9-tricosene, as a selective activator of Or7a neurons, can thereby drive strong egg-laying preferences. These new results are presented in a new Figure 8 and a new Figure 8—figure supplement 1.

C) A revised discussion should clearly acknowledge and integrate previous findings. It is puzzling that a pheromone is detected by a generalist Or. However, even if the authors do not have a good answer, it is still a surprise and therefore an issue that should be highlighted so that all olfactory biologists think about it.

We have now included additional discussion about this puzzling finding in the revised manuscript (subsection “Male Pheromones Influence a Female Oviposition Decision”). We hypothesize that 9-tricosene, since it is a selective activator of Or7a neurons, can drive a specific behavior. Other odorants that activate Or7a will also activate other odorant receptors, which will modulate or mask the behaviors driven by Or7a neurons. This might be useful to the fly in selecting oviposition sites based on different environmental and olfactory conditions.

*4) The authors emphasize that only “food odors or complex odors” could trigger the deposition of 9-tricosene. However, they did not provide sufficient experimental support to define the chemical or biological meaning of “complex food odors”. Which types of ORNs respond to the complex food odors? How many different types of ORNs are required to trigger the release of 9-tricosene? These are not trivial questions and deserve to be addressed fully in a separate study. The current manuscript will benefit significantly if the authors could remove some panels in*
Figure 1
*and*
Figure 2
*and combine these two figures to focus on the key finding that a food odor (ACV) triggers the release of an aggregation pheromone from male flies.*

We have now combined previous Figure 1 and Figure 2 by removing the suggested panels to focus on the role of ACV to trigger release of an aggregation pheromone. We have removed from the manuscript results and discussion relating to investigating how food-odors might trigger pheromone deposition. We thank the reviewers for understanding this is a complex issue, and deserves addressing in a separate study.

*5) Similarly distracting is the molecular evolution analysis shown in*
Figure 8*. They conclude their study by looking at the evolution rate of olfactory receptors in 9* Drosophila *species and find that* Or7a *is a gene that underwent many amino acid substitutions, suggesting that this receptor is one of the most variable amongst different species and thus one of the least conserved.*

*The authors seem to imply that 9-tricosene/Or7a interaction is unique for* Drosophila melanogaster*. However, without testing whether 9-tricosene also activates ab4A or triggers similar aggregation or oviposition behavior in other* Drosophila *species, their molecular evolution analysis alone is insufficient and their interpretation appears forced. If the authors feel very strongly about making this point in the current study, they will need to conduct similar electrophysiological and behavioral experiments in other* Drosophila *species. Otherwise, it may help to streamline the work by removing*
Figure 8*.*

We have removed original Figure 8 and all data/discussion pertaining to this figure.

[Editors' note: further revisions were requested prior to acceptance, as described below.]

All of the reviewers felt that the manuscript had been greatly improved in revision and gained in clarity and coherence. However, they had two concerns that need to be addressed, ideally through specific additional experiments or, if these prove technically difficult, through necessary changes to the text.

*1) Based on*
Figure 3
*and*
Figure 3*, it appears that the manuscript is incorrect in attributing the effect they observed with the pheromone blend to 9-T alone. The main reason is that the “E” distribution of the flies does not recapitulate the initial pattern and that the subsequent tests are different in essence from the initial tests.*
Figure 3
*show clear attraction, while*
Figure 3
*appears to show contact repulsion.*

Two possible reasons for this should be tested:

A) The difference could be linked to the concentration of 9-T. This could be addressed by testing different concentrations – lower or higher of 9-T. Ideally, the authors should calculate the amount of 9T in the pheromone blend used in 3B and repeat the 3F experiment with a similar 9T quantity.

*B) It is possible that there is an interplay between olfactory and gustatory modalities. Is it possible that one is olfactory (3D) and the apparent contact repulsion to 9-T is gustatory (3F)? This can be determined by testing the taste mutants (*ppk23 *and* poxn*) for their response to 9-T alone (as in 3F) as has been been done in*
Figure 2
*for the natural pheromone blend. The same experiment with* Or7a *mutants will also be informative.*

The Discussion should be modified in accordance with the results of these experiments, if necessary to acknowledge that the effect of 9-T “alone” might be different than of 9-T + 7-T (which is really the dominant chemical in this blend).

This is indeed an interesting observation, and we have planned to pursue questions pertaining to the function of *Drosophila* pheromone(s) trails in a follow-up study. I would like to first emphasize that the “E” pattern is not the primary olfactory aggregation assay we used for our aggregation experiments. The main aggregation phenotype, in which all olfactory assays were conducted, is in the 4-field arena. In the 4-field assay, we did utilize 'taste' mutants to determine the role of these mutants in aggregation to the full pheromone(s) deposited by the flies (Figure 2), as well as to 9-tricosene (Figure 5). These experiments showed that taste was not required for aggregation to the pheromone. The “E” trail assay was used mainly to verify that our hexane extracts contained functional aggregation molecules as inputs for GC-MS analyses. As such, pipetting 9-tricosene directly onto the arena in such a fashion may not closely mimic the aggregation 4-field assay in which 9-tricosene is deposited over a large surface area. Furthermore, additional focus on the “E” experiments may distract readers from the major findings of the study which demonstrate that an apple-cider vinegar odor-induced pheromone affects female oviposition decisions via Or7a receptors. For these reasons, we are hesitant to continue with technically challenging and extensive experiments on the 9-tricosene “E” pattern, preferring to instead explore these questions more rigorously and extensively in a separate study. At this point, we prefer to modify the text to address these concerns, and raise the possibility that 9-tricosene might mediate olfactory or gustatory avoidance at high concentrations.

We have highlighted these points more clearly in the revised manuscript (please see the Results subsection “9-Tricosene is a Food-Odor Induced Aggregation Pheromone”). We highlight the differences between the full hexane extract and the 9-tricosene trail. We mention that the sensory (olfactory or gustatory) cause for these differences are currently unknown, but might arise from high concentrations of 9-tricosene. We mention that these high concentrations of 9-tricosene, however, are unlikely to be found in the 4-field aggregation assays, which might suggest why such repulsion was not found in our other behavioral experiments with 9-tricosene.

We now extend the discussion of pheromone trails, which allows us to again highlight the 9-tricosene “E” observations. We mention that the cause of these differences will require future investigation, in particular how other pheromones or odorants may modulate the behavioral responses mediated by 9-tricosene alone. We again caution, however, that it remains to be determined if the high 9-tricosene concentrations causing repulsion would be found naturally in a pheromone trail.

We mention now explicitly in the figure legend for Figure 3 that the 9-tricosene pattern appears different from the full hexane extract pattern, and briefly mention potential causes for these differences.

2) The discussion on the role of the generalist Or7a could be improved and made more balanced.

We have revised the manuscript to better introduce and discuss the role of Or7a as a generalist odorant receptor, and present a more balanced view. In particular:

Abstract: We introduce the notion that Or7a is activated by many other odorants. (Space limitations to the abstract restricted more extensive changes).

Introduction: We introduce more clearly what was known about pheromone receptors, and the notion that they are narrowly tuned. This is raised to better contrast Or7a’s broader tuning, and surprise as a pheromone receptor. In addition, our summary introductory paragraph now emphasizes that Or7a was previously found to be a generalist, in contrast to previously identified pheromone receptors.

Results, subsection “The Or7a Receptor is Necessary and Sufficient for 9-tricosene Activation”: When Or7a is identified as the receptor responding to 9-tricosene, we now mention why this was surprising in terms of what was previously known about pheromone receptors.

Discussion: In the general summary paragraph, we now re-iterate the surprising finding that the pheromone 9-tricosone functions via a generalist odorant receptor. We also begin the discussion on how a generalist odorant receptor may mediate a specific pheromone response.

Figure 9 legend: We now mention that Or7a activation by other odorants may guide similar behaviors as those mediated by 9-tricosene.